# Heterogeneous Federated Fine-Tuning with Parallel One-Rank Adaptation

**Zikai Zhang   Rui Hu** * **Jiahao Xu**
University of Nevada, Reno, USA
{zikaiz, ruihu, jiahaox}@unr.edu

## Abstract

Large Language Models (LLMs) have demonstrated remarkable effectiveness in adapting to downstream tasks through fine-tuning. Federated Learning (FL) extends this capability by enabling collaborative fine-tuning across distributed clients using Low-Rank Adaptation (LoRA), while preserving data privacy by avoiding raw data sharing. However, practical deployments face challenges when clients have heterogeneous resources and thus adopt different LoRA ranks, leading to substantial initialization and aggregation noise that undermines performance. To address these challenges, we propose Fed-PLoRA, a novel lightweight heterogeneous federated fine-tuning (FFT) framework. Fed-PLoRA introduces Parallel One-Rank Adaptation (PLoRA), a new LoRA variant that replaces the classic multi-rank LoRA module with multiple parallel one-rank modules, and a novel Select-N-Fold strategy that folds untrained PLoRA modules into the pre-trained weights before local training, thereby accommodating heterogeneous client resources. We provide a unified analysis of initialization and aggregation noise of Fed-PLoRA and demonstrate how it addresses the limitations of state-of-the-art methods. Extensive experiments on diverse LLM fine-tuning tasks demonstrate that Fed-PLoRA consistently outperforms existing methods in both accuracy and efficiency. The code is available at `https://github.com/TNI-playground/Fed-PLoRA`.

## 1 Introduction

Federated Learning (FL) has emerged as a crucial paradigm for fine-tuning Large Language Models (LLMs) across multiple clients while preserving data privacy by avoiding raw data sharing. Low-Rank Adaptation (LoRA) Hu et al. (2021) is a widely used parameter-efficient fine-tuning (PEFT) method, which adapts LLMs by injecting trainable low-rank matrices into pretrained weight spaces. Specifically, LoRA factorizes a weight update matrix $\Delta W \in \mathbb{R}^{d \times k}$ as $\Delta W = \mathbf{BA}$, $\mathbf{B} \in \mathbb{R}^{d \times r}$, $\mathbf{A} \in \mathbb{R}^{r \times k}$, where $r \ll \min(d, k)$ denoting the rank. Combining FL and LoRA provides an effective framework for collaborative LLM fine-tuning on downstream tasks Wang et al. (2018). A representative example FedIT Zhang et al. (2023), which extends the classical Federated Averaging (FedAvg) McMahan et al. (2017) algorithm to the LoRA setting. In FedIT, each client $i$ locally trains its LoRA modules $(\mathbf{A}_i, \mathbf{B}_i)$ while keeping the pretrained backbone frozen. At the end of each round, the client uploads its LoRA matrices to the server. The server then aggregates them by weighted averaging (i.e., $\mathbf{A} = \sum \omega_i \mathbf{A}_i$ and $\mathbf{B} = \omega_i \mathbf{B}_i$ with weights $\omega_i$). The resulting global LoRA modules are distributed back to all clients and serve as the initialization for the next training round. This iterative process continues until convergence of the global adapted model.

However, client resource heterogeneity poses a major challenge for federated LoRA-based fine-tuning. In practice, clients differ significantly in computational capacity, which directly constrains the LoRA ranks they can afford to train Wang et al. (2024); Bai et al. (2024). Prior works Haobo et al. (2024); Ren et al. (2024) demonstrate that the LoRA rank critically influences both fine-tuning performance and resource consumption. Higher ranks generally enhance adaptation capacity, but at the cost of greater computational and communication overhead. As a result, resource-constrained clients are often unable to adopt the larger ranks that more capable clients can sustain. This im-

---

*Corresponding author.

balance undermines global aggregation, leading to degraded model performance and limiting the effective participation of weaker clients. These challenges highlight the urgent need for methods that can accommodate heterogeneous LoRA ranks across clients, thereby enabling effective and inclusive federated fine-tuning of LLMs.

Existing studies Wang et al. (2024); Cho et al. (2024); Bai et al. (2024); Zhang et al. (2024b; 2025) on federated fine-tuning (FFT) with heterogeneous LoRA ranks primarily address the challenge of aggregating local LoRA modules with varying dimensions. For example, FLoRA Wang et al. (2024) proposes a stacking-based aggregation scheme, where local LoRA modules are concatenated to construct global modules, thereby accommodating heterogeneous ranks during aggregation. Beyond aggregation, local initialization poses another key challenge in heterogeneous-rank FFT. Since the global aggregated module may have a different rank from that of clients, mismatches naturally arise. For instance, HETLoRA Cho et al. (2024) initializes local modules by truncating the global LoRA matrices from lower to higher rank indices according to each client's rank, whereas FLoRA resorts to random re-initialization in each round. Both strategies introduce substantial initialization noise, in contrast to homogeneous settings (e.g., FedIT), where local modules can be directly initialized with the global ones of identical rank.

In this paper, we provide a unified analysis of initialization and aggregation noise across several representative methods for heterogeneous FFT. Building on these insights, we propose a novel heterogeneous FFT framework, **Fed-PLoRA**, which incorporates *Parallel One-Rank Adaptation (PLoRA)*. Unlike the classical LoRA approach that relies on a single multi-rank matrix, PLoRA constructs modules of any desired rank by combining multiple parallel one-rank matrices. This design naturally supports client resource heterogeneity through a novel *Select-N-Fold* strategy. Our theoretical results show that Fed-PLoRA achieves near-optimal local initialization while minimizing aggregation noise, thereby enabling effective and inclusive fine-tuning in heterogeneous federated settings. Our main contributions are summarized as follows:

- We present a unified analysis of initialization and aggregation noise in heterogeneous FFT with varying LoRA ranks. Guided by this analysis, we propose Fed-PLoRA, a lightweight framework that naturally accommodates heterogeneous client capacities while ensuring stable initialization and low aggregation noise. Fed-PLoRA is lightweight and can be seamlessly integrated into existing LoRA and FL pipelines.

- Fed-PLoRA incorporates PLoRA, a new LoRA variant that substitutes a single multi-rank LoRA module with multiple parallel one-rank modules. Building on this design, we propose the Select-N-Fold strategy, where each client trains only a subset of PLoRA modules according to its computational budget, while folding the remaining modules into the frozen pretrained weights.

- Through extensive experiments on diverse LLM fine-tuning tasks, we demonstrate that Fed-PLoRA consistently outperforms existing heterogeneous LoRA-based FFT methods across varying client resource and data settings.

## 2 FEDERATED FINE-TUNING SYSTEM

We consider an FFT system consisting of a central server and $v$ clients. The server maintains a global transformer-based pre-trained LLM, denoted by $\Theta^0$, while each client $i \in [v]$ owns a local dataset $D_i$. For LoRA-based FFT, LoRA can be applied to a pre-trained target module $\mathcal{W}^0 \in \mathbb{R}^{d \times k}$ (e.g., a query projection) within a transformer block of $\Theta^0$.

LoRA constrains the update of the target module, $\Delta \mathcal{W}$, through a low-rank factorization such that $\mathcal{W}^0 + \Delta \mathcal{W} = \mathcal{W}^0 + \mathbf{BA}$, where $\mathbf{A} \in \mathbb{R}^{r \times k}$, $\mathbf{B} \in \mathbb{R}^{d \times r}$, with the rank $r \ll \min(d, k)$. In practice, LoRA is applied to $L$ distinct target modules within $\Theta^0$, denoted by $\mathcal{W}^0 := \{\mathcal{W}^{0,l}\}_{l=1}^{L}$. We define the corresponding collection of LoRA parameters as $\theta := \{\mathbf{A}, \mathbf{B}\}$ with $\mathbf{A} = \{\mathbf{A}^l\}_{l=1}^{L}$ and $\mathbf{B} = \{\mathbf{B}^l\}_{l=1}^{L}$, and each pair $\{\mathbf{A}^l, \mathbf{B}^l\}$ represents the low-rank matrices associated with the $l$-th target module. Given this setup, the problem of FFT can be formulated as follows:

$$\min_{\theta} \mathcal{L}(\theta) := \frac{1}{v} \sum_{i=1}^{v} \mathcal{L}_i(\theta; \Theta^0), \tag{1}$$

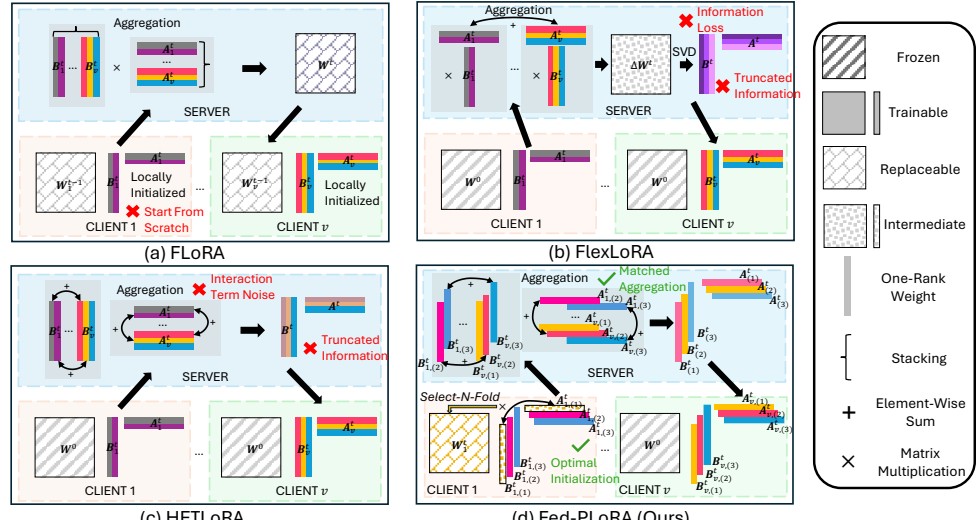

Figure 1: Framework of FLoRA, FlexLoRA, HETLoRA, and our method Fed-PLoRA.

where $\mathcal{L}_i(\boldsymbol{\theta}; \boldsymbol{\Theta}^0) := \mathbb{E}_{x \in D_i}[\ell(\boldsymbol{\theta}; \boldsymbol{\Theta}^0, x)]$ denotes the local objective function of client $i$, and $\ell(\boldsymbol{\theta}; \boldsymbol{\Theta}^0, x)$ is the loss of the model on a datapoint $x$ sampled from $D_i$.

In a heterogeneous rank setting, each client $i$ employs LoRA modules with its own LoRA rank $r_i$. To solve the optimization problem in Equation equation 1, FFT methods Cho et al. (2024); Wang et al. (2024); Bai et al. (2024) typically follow a three-step procedure in each training round $t \in [T]$:

**1. Broadcast and Initialization:** The server broadcasts the global model to clients, and client $i$ initializes its local LoRA modules with rank $r_i$ via $\boldsymbol{\theta}_i^{t-1} = \mathsf{Init}(\boldsymbol{\theta}^{t-1}, r_i)$.

**2. Local Client Update:** Each client $i$ then fine-tunes its initialized LoRA parameters $\boldsymbol{\theta}_i^{t-1}$ using its local dataset $D_i$, while keeping the pretrained backbone $\boldsymbol{\Theta}^0$ frozen. This process typically involves multiple steps of Stochastic Gradient Descent (SGD) or its variants Kingma (2014), resulting in updated local LoRA parameters: $\boldsymbol{\theta}_i^t = \mathsf{LocalUpdate}(\boldsymbol{\theta}_i^{t-1}, D_i, \boldsymbol{\Theta}^0)$. The updated parameters $\boldsymbol{\theta}_i^t$ are then transmitted back to the server.

**3. Server-Side Aggregation:** The server collects the set of updated parameters $\{\boldsymbol{\theta}_i^t\}_{i \in [v]}$ and aggregates them to form the global LoRA parameters $\boldsymbol{\theta}^t$ for round $t$: $\boldsymbol{\theta}^t = \{\mathbf{A}^t, \mathbf{B}^t\} = \mathsf{Agg}(\{\boldsymbol{\theta}_i^t\}_{i \in [v]})$, where $\mathbf{A}^t \in \mathbb{R}^{R \times k}$ and $\mathbf{B}^t \in \mathbb{R}^{d \times R}$. Here, the aggregation operator $\mathsf{Agg}(\cdot)$ must be capable of combining local LoRA modules with heterogeneous ranks, and the resulting global rank $R$, which satisfies $R \geq \max(r_i)$, is determined by the choice of aggregation rule.

# 3 PARALLEL ONE-RANK ADAPTATION FOR HETEROGENEOUS FFT

In this section, we introduce Fed-PLoRA, a heterogeneous FFT framework that incorporates a new LoRA variant (PLoRA) and a novel local initialization strategy (Select-N-Fold) to eliminate initialization noise and mitigate aggregation noise.

## 3.1 MOTIVATION

Existing LoRA-based heterogeneous FFT methods Cho et al. (2024); Bai et al. (2024); Wang et al. (2024) that follow the above training framework differ primarily in their initialization and aggregation strategies. To analyze their effects, we formally define the notions of initialization noise and aggregation noise as follows:

*Initialization Noise:* During the initialization step of round $t$, clients with limited local rank $r_i$ may be unable to fully accommodate the information contained in the global LoRA modules as $r_i \leq R$.

This mismatch introduces initialization noise, defined as the total initialization gap across all clients:

$$\mathcal{N}_{\text{Init}}^t := \sum_{i \in [v]} \left( \left\| \mathbf{A}^{t-1} \ominus \mathbf{A}_i^{t-1} \right\|_F + \left\| \mathbf{B}^{t-1} \ominus \mathbf{B}_i^{t-1} \right\|_F \right), \tag{2}$$

where $\|\cdot\|_F$ denotes the Frobenius norm, and the operator $\ominus$ represents a rank-wise subtraction. This metric quantifies the information that resource-constrained clients fail to retain when initializing from the global model. In homogeneous rank settings (i.e., $r_i = R$ for all $i \in [v]$), the initialization noise vanishes since all clients can directly adopt the global LoRA modules.

*Aggregation noise:* After local training, the server receives updated parameters $\boldsymbol{\theta}_i^t$ from each client $i$. Following prior works Wang et al. (2024); Sun et al. (2024), we define an *ideal* model update $\Delta \mathcal{W}_*^t$ as the average of the local model updates, and the deviation from this ideal model update defines the aggregation noise, i.e.,

$$\mathcal{N}_{\text{Agg}}^t := \left\| \Delta \mathcal{W}_*^t - \Delta \mathcal{W}^t \right\|_F, \text{ with } \Delta \mathcal{W}_*^t = \frac{1}{v} \sum_{i \in [v]} \Delta \mathcal{W}_i^t = \frac{1}{v} \sum_{i \in [v]} \mathbf{B}_i^t \mathbf{A}_i^t, \tag{3}$$

where $\Delta \mathcal{W}^t = \mathbf{B}^t \mathbf{A}^t$ represents the actual model update obtained by a specific aggregation method. Ideally, a perfect aggregation method yields $\mathcal{N}_{\text{Agg}}^t = 0$.

## 3.2 PARALLEL ONE-RANK ADAPTATION

Before presenting the Fed-PLoRA framework, we first introduce its fundamental building block, PLoRA. The key idea is to replace a single multi-rank LoRA module with multiple parallel one-rank modules. Concretely, consider applying a LoRA module with rank $R$ to a target weight matrix $\mathcal{W}^0 \in \mathbb{R}^{d \times k}$. In classical LoRA, this is parameterized by matrices $\mathbf{A} \in \mathbb{R}^{R \times k}$ and $\mathbf{B} \in \mathbb{R}^{d \times R}$, yielding the update $\Delta \mathcal{W}_{\text{LoRA}} := \mathbf{B} \mathbf{A}$. In contrast, PLoRA decomposes this rank-$R$ module into $R$ parallel one-rank components. For each $j \in [R]$, a PLoRA component consists of a pair of one-rank matrices $\mathbf{A}_{(j)} \in \mathbb{R}^{1 \times k}$ and $\mathbf{B}_{(j)} \in \mathbb{R}^{d \times 1}$ (see Figure 1(d)), producing an individual update $\Delta \mathcal{W}_{(j)} = \mathbf{B}_{(j)} \mathbf{A}_{(j)}$. The overall update is simply the sum of these contributions:

$$\Delta \mathcal{W}_{\text{PLoRA}} := \sum_{j=1}^R \Delta \mathcal{W}_{(j)} = \sum_{j=1}^R \mathbf{B}_{(j)} \mathbf{A}_{(j)} = \sum_{j=1}^R \mathbf{B}_{[:,j]} \mathbf{A}_{[j,:]} = \Delta \mathcal{W}_{\text{LoRA}}$$

which is mathematically equivalent to the classic multi-rank LoRA formulation, since $\mathbf{B}(j) = \mathbf{B}[:,j]$ and $\mathbf{A}(j) = \mathbf{A}[j,:]$. Thus, PLoRA achieves the same adaptation effect and parameter efficiency as standard multi-rank LoRA, while enabling a modular decomposition that is naturally suited to heterogeneous FTT.

## 3.3 FED-PLoRA: HETEROGENEOUS FFT WITH PLoRA

As in existing approaches, resource-constrained clients can reduce their LoRA ranks to fit limited computational budgets, which naturally results in heterogeneous ranks across clients. Within our framework, this corresponds to adjusting the number of parallel one-rank matrix pairs in the PLoRA modules. However, this adjustment alone does not eliminate initialization and aggregation noise.

To overcome these issues in heterogeneous settings, we propose a novel *Select-N-Fold* strategy within the Fed-PLoRA framework. This strategy is specifically designed to handle rank heterogeneity, ensuring zero initialization noise and minimizing aggregation noise. Assume the server maintains global PLoRA parameters $\boldsymbol{\theta} = \{\{\mathbf{A}^l\}_{l \in [L]}, \{\mathbf{B}^l\}_{l \in [L]}\}$, where for each target module $l \in [L]$, $\mathbf{A}^l := \{\mathbf{A}_{(j)}^l\}_{j \in [R]}$ and $\mathbf{B}^l := \{\mathbf{B}_{(j)}^l\}_{j \in [R]}$ represent the $R$ parallel one-rank PLoRA components. For client $i$, the local PLoRA parameters are $\boldsymbol{\theta}_i = \{\{\mathbf{A}_i^l\}_{l \in [L]}, \{\mathbf{B}_i^l\}_{l \in [L]}\}$, where $\mathbf{A}_i^l := \{\mathbf{A}_{i,(j)}^l\}_{j \in [r_i]}$ and $\mathbf{B}_i^l := \{\mathbf{B}_{i,(j)}^l\}_{j \in [r_i]}$ with local rank $r_i$. The global rank is chosen such that $R \geq \max_{i \in [v]} r_i$. Fed-PLoRA follows the three-step training framework described in Section 2, and the pseudo-code is shown in Algorithm 1. In each training round $t$:

**Broadcast and Initialization:** The server broadcasts the global PLoRA parameters $\boldsymbol{\theta}^{t-1}$ to clients. Each client $i$ then randomly selects a subset $\mathcal{K}_i^t$ of $r_i$ parallel one-rank PloRA modules for local

training. For clarity, we omit the index $l$ in the notation, but note that this selection and the subsequence operations are performed independently for each target module $l \in [L]$. The local PLoRA parameters of client $i$ are initialized as $\boldsymbol{\theta}_i^{t-1} = \{\mathbf{A}_{(j)}^{t-1}, \mathbf{B}_{(j)}^{t-1}\}_{j \in \mathcal{K}_i^t}$. The remaining $(R - r_i)$ parallel one-rank PLoRA modules are temporarily **folded** into the corresponding pre-trained target module $\boldsymbol{\mathcal{W}}^0$, yielding the local target module

$$\boldsymbol{\mathcal{W}}_i^t := \boldsymbol{\mathcal{W}}^0 + \Delta\boldsymbol{\mathcal{W}}_{\text{fold},i}^{t-1} = \boldsymbol{\mathcal{W}}^0 + \sum_{j \notin \mathcal{K}_i^t} \mathbf{B}_{(j)}^{t-1} \mathbf{A}_{(j)}^{t-1},$$

which then remains frozen during local training. This procedure allows the client to preserve information from the full set of global PloRA modules while training only a subset. Consequently, client $i$ operates with an effective rank of $r_i$, matching the parameter count of a standard rank-$r_i$ LoRA configuration, but without incurring initialization noise.

*Randomness in Select-N-Fold:* Our objective is to obtain a well-trained global PLoRA that can be applied to the pretrained backbone for downstream tasks. When local resource is sufficient (i.e., $r_i = R$), all local PLoRA modules can be trained, ensuring that every global PLoRA module is updated. When $r_i < R$, trainable modules are selected independently across clients and target modules; hence, in expectation, each global module is updated by some subset of clients in every round. This randomness mitigates the risk of long-term staleness in global PLoRA modules under the Select-N-Fold strategy.

**Local Client Update:** During local training, each client $i$ updates only the selected PLoRA modules in $\mathcal{K}_i^t$. After optimization, the updated local PLoRA parameters are $\boldsymbol{\theta}_i^t = \{\mathbf{A}_{i,(j)}^t, \mathbf{B}_{i,(j)}^t\}_{j \in \mathcal{K}_i^t}$, which are then transmitted to the server.

**Server-Side Aggregation:** The server aggregates these local PLoRA modules in a rank-wise manner. For each $j \in [R]$, the global PLoRA modules are computed as $\mathbf{A}_{(j)}^t = \frac{1}{|\mathcal{Q}_{(j)}^t|} \sum_{i \in \mathcal{Q}_{(j)}^t} \mathbf{A}_{i,(j)}^t$, $\mathbf{B}_{(j)}^t = \frac{1}{|\mathcal{Q}_{(j)}^t|} \sum_{i \in \mathcal{Q}_{(j)}^t} \mathbf{B}_{i,(j)}^t$, where $\mathcal{Q}_{(j)}^t = \{i | i \in [v], j \in \mathcal{K}_i^t\}$ denotes the set of clients that updated module $j$ in round $t$. The aggregated global PLoRA parameters $\boldsymbol{\theta}^t = \{\mathbf{A}^t, \mathbf{B}^t\}$ are then used for the next round of training. The process iterates until the global model converges.

### 3.4 ANALYSIS OF INITIALIZATION AND AGGREGATION NOISE

Here, we analyze the initialization and aggregation noise of Fed-PLoRA, comparing it with three SOTA methods, FLoRA Wang et al. (2024), FlexLoRA Bai et al. (2024), and HETLoRA Cho et al. (2024). Due to page limitations, we provide the detailed derivation in Appendix E.

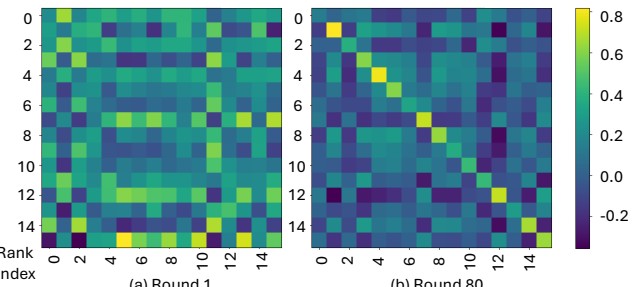

(a) Round 1                    (b) Round 80

Figure 2: Cosine similarities between parallel one-rank modules of PLoRA. (a) Low, random similarities at initialization; (b) Increasing within-rank similarity across clients, while cross-rank similarity remains low, indicating that modules capture distinct knowledge at different ranks but converge on similar knowledge across clients, despite differences in resource limitations and data distributions (see *Appendix Section C.3*).

**Fed-PLoRA.** Because each client either trains or folds the globally initialized one-rank modules into its frozen target weights, there is no discrepancy between local and global modules. Hence, $\mathcal{N}_{\text{Ours\_Init}}^t = 0$. Under our aggregation rule, the aggregation noise is $\mathcal{N}_{\text{Ours\_Agg}}^t = \| \sum_{j=1}^R ((1/|\mathcal{Q}_{(j)}^t|) \sum_{i \in \mathcal{Q}_{(j)}^t} (\mathbf{B}_{i,(j)}^t - \overline{\mathbf{B}}_{(j)}^t)(\mathbf{A}_{i,(j)}^t - \overline{\mathbf{A}}_{(j)}^t)) \|_F$, where $\overline{\mathbf{A}}_{(j)}^t := (1/|\mathcal{Q}_{(j)}^t|) \sum_{i \in \mathcal{Q}_{(j)}^t} \mathbf{A}_{i,(j)}^t$ and $\overline{\mathbf{B}}_{(j)}^t := (1/|\mathcal{Q}_{(j)}^t|) \sum_{i \in \mathcal{Q}_{(j)}^t} \mathbf{B}_{i,(j)}^t$. This noise vanishes when the cross-client covariance between $\mathbf{A}_{i,(j)}^t$ and $\mathbf{B}_{i,(j)}^t$ is zero for every rank. More generally, by the Cauchy–Schwarz inequality, it can be bounded as $\mathcal{N}_{\text{Ours\_Agg}}^t \leq \sum_{j=1}^R \frac{1}{|\mathcal{Q}_{(j)}^t|} \sum_{i \in \mathcal{Q}_{(j)}^t} \|\mathbf{B}_{i,(j)}^t - \overline{\mathbf{B}}_{(j)}^t\|_2 + \|\mathbf{A}_{i,(j)}^t - \overline{\mathbf{A}}_{(j)}^t\|_2$. As shown in the

heatmap in Figure 2, the diagonal entries represent the similarity of the $j$-th PLoRA module ($j \in [R]$) across clients, averaged over all client pairs. We observe that these similarities increase significantly as training progresses, indicating that local PLoRA modules become more aligned across clients. This alignment reduces deviations from the mean, thereby tightening the upper bound of the aggregation noise.

**FLoRA.** As shown in Figure 1 (a), FLoRA employs a stacking-based aggregation method to accommodate LoRA modules from clients with heterogeneous ranks. Each client $i$ initializes its trainable LoRA modules $\{\mathbf{A}_i^{t-1}, \mathbf{B}_i^{t-1}\}$ with rank $r_i$ from scratch (e.g., $\mathbf{A}_i^{t-1}$ drawn from a normal distribution and $\mathbf{B}_i^{t-1}$ set to zeros). Before local training, the local target module is replaced with the latest global target module from the server. After local training, the updated LoRA modules $\{\mathbf{A}_i^t, \mathbf{B}_i^t\}$ are sent to the server, which concatenates all client updates along the rank dimension to form two large intermediate matrices: $\{\mathbf{A}^t \in \mathbb{R}^{(\sum_{i \in [v]} r_i) \times k}, \mathbf{B}^t \in \mathbb{R}^{d \times (\sum_{i \in [v]} r_i)}\}$. The global update is then computed as $\Delta \mathcal{W}_{\text{FLoRA}}^t = \mathbf{B}^t \mathbf{A}^t / v$. Because local LoRA modules $(\mathbf{A}_i^{t-1}, \mathbf{B}_i^{t-1})$ are freshly initialized each round, the initialization noise is $\mathcal{N}_{\text{FLoRA\_Init}}^t = \sum_{i \in [v]} \left( \left\| \mathbf{A}^{t-1} \right\|_F + \left\| \mathbf{B}^{t-1} \right\|_F \right) + \sigma^2$, where $\sigma^2$ reflects the variance of the random initialization for $\mathbf{A}$. This noise can be substantial and may degrade local training performance. On the other hand, the stacking operation in FLoRA preserves the local updates without distortion, leading to zero aggregation noise, i.e., $\mathcal{N}_{\text{FLoRA\_Agg}}^t = 0$.

**FlexLoRA.** As shown in Figure 1 (b), FlexLoRA applies singular value decomposition (SVD) to the global target module update $\Delta \mathcal{W}^{t-1}$ to construct global LoRA modules $\{\mathbf{A}^{t-1}, \mathbf{B}^{t-1}\}$ with rank $R$, i.e., $\Delta \mathcal{W}^{t-1} \approx \mathbf{B}_{\text{svd}}^{t-1} \mathbf{A}_{\text{svd}}^{t-1}$ where $\mathbf{B}_{\text{svd}}^{t-1} := \mathbf{U}^{t-1} \mathbf{S}^{t-1}$ and $\mathbf{A}_{\text{svd}}^{t-1} := \mathbf{V}^{t-1 \top}$. Here, $\mathbf{U}^{t-1} \in \mathbb{R}^{d \times R}$ contains the top-$R$ left singular vectors, $\mathbf{S}^{t-1} \in \mathbb{R}^{R \times R}$ is the diagonal matrix of singular values, and $\mathbf{V}^{t-1} \in \mathbb{R}^{R \times k}$ holds the top-$R$ right singular vectors. For a client with rank $r_i$, the local LoRA modules are initialized by truncating to the top $r_i$ components: $\mathbf{A}_i^{t-1} = \mathbf{V}_{[:r_i,:]}^{t-1 \top}$, $\mathbf{B}_i^{t-1} = \mathbf{U}_{[:,:r_i]}^{t-1} \mathbf{S}_{[:r_i,:r_i]}^{t-1}$. After local fine-tuning, the server aggregates client updates to form the global target module update: $\Delta \mathcal{W}^t = \frac{1}{v} \sum_{i \in [v]} \Delta \mathcal{W}_i^t = \frac{1}{v} \sum_{i \in [v]} \mathbf{B}_i^t \mathbf{A}_i^t$. Because each client only retains the top-$r_i$ singular components, the initialization noise is $\mathcal{N}_{\text{FlexLoRA\_Init}}^t = \sum_{i \in [v]} (\|\mathbf{A}_{[r_i+1:R,:]}^{t-1}\|_F + \|\mathbf{B}_{[:,r_i+1:R]}^{t-1}\|_F)$ which increases as $r_i$ decreases. The server obtains the exact average update $\Delta \mathcal{W}^t$, but when reconstructing global LoRA modules of rank $R$ via SVD, decomposition error is introduced: $\mathcal{N}_{\text{FlexLoRA\_Agg}}^t = \|\Delta \mathcal{W}^t - \mathbf{U}^t \mathbf{S}^t \mathbf{V}^{t \top}\|_F$. This error increases as global rank $R$ increases.

**HETLoRA.** As shown in Figure 1 (c), HETLoRA initializes each client's LoRA modules by truncating the global LoRA modules, i.e., $\mathbf{A}_i^{t-1} := \mathbf{A}_{[:r_i,:]}^{t-1}, \mathbf{B}_i^{t-1} := \mathbf{B}_{[:,:r_i]}^{t-1}$. After local fine-tuning, the updated modules $\{\mathbf{A}_i^t, \mathbf{B}_i^t\}$ are sent to the server, where they are expanded to rank $R$ by zero-padding. The padded modules $\mathbf{A}_i^{t,\prime}, \mathbf{B}_i^{t,\prime}$ are then averaged to form the global LoRA modules: $\mathbf{A}^t = \frac{1}{v} \sum_{i \in [v]} \mathbf{A}_i^{t,\prime}, \mathbf{B}^t = \frac{1}{v} \sum_{i \in [v]} \mathbf{B}_i^{t,\prime}$. Because each client discards the bottom $(R - r_i)$ components, the initialization noise is $\mathcal{N}_{\text{HETLoRA\_Init}}^t = \sum_{i \in [v]} (\|\mathbf{A}_{[r_i+1:R,:]}^{t-1}\|_F + \|\mathbf{B}_{[:,r_i+1:R]}^{t-1}\|_F)$, which grows as $r_i$ decreases. For aggregation, the global update is computed as $\Delta \mathcal{W}^t = \mathbf{B}^t \mathbf{A}^t$, but because $\mathbf{A}^t$ and $\mathbf{B}^t$ are averaged separately, this introduces a mathematical bias relative to the optimal update $\frac{1}{v} \sum_i \mathbf{B}_i^{t,\prime} \mathbf{A}_i^{t,\prime}$ Wang et al. (2024); Sun et al. (2024). The resulting aggregation noise is $\mathcal{N}_{\text{HETLoRA\_Agg}}^t = \|(1/v^2)((v-1) \sum_{i \in [v]} \mathbf{B}_i^{t,\prime} \mathbf{A}_i^{t,\prime} - \sum_{j \in [v]} \sum_{k \in [v], k \neq j} \mathbf{B}_j^{t,\prime} \mathbf{A}_k^{t,\prime})\|_F$.

In summary, FLoRA suffers from substantial initialization noise because each client's LoRA modules are randomly re-initialized. FlexLoRA and HETLoRA reduce this randomness by truncating global modules, but still incur initialization noise that grows as more clients operate with smaller ranks. Moreover, HETLoRA introduces structural aggregation bias, while FlexLoRA suffers decomposition error from SVD. In contrast, Fed-PLoRA eliminates initialization noise entirely and minimizes aggregation noise by integrating PLoRA with the Select-N-Fold strategy, leading to more stable and accurate FFT under heterogeneous client capacities.

## 3.5 COMMUNICATION, COMPUTATION, AND MEMORY OVERHEAD

We analyze the communication, computation, and memory overhead of Fed-PLoRA in comparison with FLoRA, FlexLoRA, and HETLoRA.

**Communication.** In each round, every client in Fed-PLoRA uploads its local PLoRA update with rank $r_i$ to the server. The uplink payload scales as $\mathcal{O}((d+k)r_i)$, which is identical to the uplink cost of FLoRA, FlexLoRA, and HETLoRA. Thus, Fed-PLoRA introduces no additional uplink overhead. After aggregation, the server sends the global PLoRA module with rank $R$ to every client, giving a downlink cost of $\mathcal{O}((d+k)R)$. In comparison, HETLoRA and FlexLoRA send each client a personalized global LoRA module of rank $r_i$, resulting in per-client downlink cost $\mathcal{O}((d+k)r_i)$. FLoRA incurs a larger downlink cost of $\mathcal{O}(dk)$ as its server sends the updated target module. Thus, the downlink cost of Fed-PLoRA is higher than HETLoRA and FlexLoRA by $\mathcal{O}((d+k)(R-r_i))$ per client, but it reduces the downlink cost relative to FLoRA by $\mathcal{O}(dk-(d+k)R)$.

**Computation.** All methods use an identical local fine-tuning procedure with local rank $r_i$ for client $i$ after initialization, so Fed-PLoRA incurs no additional model training cost on the client side. The only difference in local computation arises during model initialization, whose cost is negligible compared with model training cost. For completeness, we also discuss the initialization cost of Fed-PLoRA compared with other methods. During initialization, all methods update their local PLoRA/LoRA parameters, either by using the received global PLoRA/LoRA module or by randomly initializing them. Compared to FlexLoRA and HETLoRA, Fed-PLoRA and FLoRA require an additional step. In Fed-PLoRA, client $i$ folds the remaining $R-r_i$ global one-rank PLoRA modules into the frozen model weights, which incurs an extra computational cost of $\mathcal{O}(dk(R-r_i))$. In FLoRA, every client updates its local target module using the received global target module, resulting in an additional computational cost of $\mathcal{O}(dk)$. On the server side, Fed-PLoRA performs rank-wise averaging directly on the received local updates, which is lightweight. In HETLoRA, the server first expands each local update to rank $R$ before averaging, introducing slightly more computation than simple rank-wise averaging. FlexLoRA requires an SVD operation to obtain a global low-rank update, which is computationally expensive. FLoRA concatenates all local updates and computes a full update to the global target module, which also incurs substantial cost. Thus, Fed-PLoRA introduces no additional server-side computational overhead compared with the other methods and is in fact the most lightweight among them.

**Memory.** During local model training, Fed-PLoRA has the same memory footprint for storing model parameters, optimizer states, activations, and other training-related tensors as HETLoRA, FLoRA, and FlexLoRA. This is because all methods follow the same local fine-tuning procedure to update only the LoRA/PLoRA module on top of the frozen backbone. The only difference in memory usage arises during model initialization, whose cost is negligible compared to the overall fine-tuning memory footprint. During local initialization, each client requires a small temporary memory buffer to hold the parameters received from the server. These parameters are immediately discarded once initialization completes, resulting in no persistent memory cost. Specifically, in Fed-PLoRA, client $i$ temporarily stores the global PLoRA module of rank $R$. In HETLoRA and FlexLoRA, client $i$ instead stores the global LoRA module of rank $r_i$. In contrast, FLoRA requires client $i$ to temporarily store the full global target module, whose size scales as $\mathcal{O}(dk)$. Thus, Fed-PLoRA incurs a temporary memory overhead of $\mathcal{O}((d+k)(R-r_i))$ compared with HETLoRA and FlexLoRA, but reduces temporary memory usage relative to FLoRA by $\mathcal{O}(dk-(d+k)R)$.

Overall, Fed-PLoRA introduce negligible overhead in communication, computation, and memory. Detailed numerical results and measurements are provided in Appendix F.2.

## 4 EVALUATION

**Models, Datasets, Baselines, and Experimental Settings.** We employ six models with different scales in our experiments: BERT-base Devlin et al. (2019), Llama-1B Chen et al. (2024), Llama-3.1-8B Meta AI (2024), OPT-1.3B Zhang et al. (2022a), Qwen3-4B-A3B-Instruct-2507 Team (2025), and Mistral-7B-v0.3 MistralAI (2023). Following the configurations in the original LoRA paper Hu et al. (2021), the LoRA modules are applied to the self-attention layers only. We use datasets across multiple domains. For general instruction following, we use Natural Instructions Wang et al. (2022) and Dolly-15K Conover et al. (2023) for training and evaluation. For general natural language understanding, we adopt the GLUE benchmark Wang et al. (2018). In the finance domain, we train on FinGPT FinGPT (2023) and evaluate on FPB Malo et al. (2014), FIQA Maia et al. (2018), and TFNS Neural Magic (2022). In the medical domain, we train on MedAlpaca Han et al. (2023) and evaluate on PubMedQA Jin et al. (2019), MedMCQA Pal et al. (2022), MedQA Jin et al. (2021), and

| Method | | Natural Instructions (IID) | Natural Instructions (non-IID) |
|---|---|---|---|
| Untuned Model | | $33.82^{+0.25}_{-0.12}$ | |
| **Homogeneous** | FedIT (Rank=$R$) | $66.88^{+0.59}_{-0.33}$ | $61.28^{+0.67}_{-0.41}$ |
| | FedIT (Rank=$R/2$) | $64.52^{+0.11}_{-0.21}$ | $60.57^{+0.08}_{-0.21}$ |
| **Heterogeneous** $R=16$ Avg Rank=8 | FLoRA | $33.82^{+0}_{-0}$ | $33.82^{+0}_{-0}$ |
| | FlexLoRA | $59.07^{+2.63}_{-3.24}$ | $53.51^{+5.27}_{-9.30}$ |
| | HETLoRA | $58.07^{+2.02}_{-1.35}$ | $58.84^{+0.32}_{-0.22}$ |
| | **Fed-PLoRA** | $\mathbf{64.96^{+1.63}_{-1.01}}$ | $\mathbf{60.76^{+0.68}_{-0.46}}$ |

Table 1: Comparison of Fed-PLoRA with baselines on IID and non-IID Natural Instructions dataset. FedIT as the baseline for FFT under homogeneous settings without resource constraints.

| Method | | CoLA | SST-2 | MRPC | QQP | QNLI | RTE | Avg |
|---|---|---|---|---|---|---|---|---|
| Untuned Model | | $1.20^{+0.10}_{-0.05}$ | $49.08^{+0.42}_{-0.37}$ | $31.61^{+0.83}_{-0.55}$ | $63.09^{+0.28}_{-0.31}$ | $49.95^{+0.67}_{-0.52}$ | $52.70^{+0.36}_{-0.44}$ | $41.27^{+0.44}_{-0.37}$ |
| **Homo.** | FedIT (Rank=$R$) | $61.68^{+1.26}_{-0.88}$ | $92.47^{+0.19}_{-0.16}$ | $87.58^{+0.41}_{-0.33}$ | $86.86^{+0.04}_{-0.02}$ | $89.53^{+0.84}_{-0.54}$ | $68.35^{+0.24}_{-0.12}$ | $81.08^{+0.50}_{-0.33}$ |
| | FedIT (Rank=$R/2$) | $59.05^{+2.27}_{-2.63}$ | $92.23^{+0.08}_{-0.15}$ | $86.16^{+0.74}_{-0.49}$ | $86.47^{+0.02}_{-0.03}$ | $88.32^{+1.04}_{-0.77}$ | $65.23^{+0}_{-0}$ | $79.58^{+0.69}_{-0.68}$ |
| **Hete.** $R=16$ Avg Rank=7 | FLoRA | $-4.08^{+2.01}_{-2.42}$ | $91.70^{+0.38}_{-0.19}$ | $43.87^{+24.51}_{-12.26}$ | $53.88^{+8.79}_{-15.13}$ | $50.47^{+2.94}_{-3.10}$ | $51.02^{+2.04}_{-3.73}$ | $47.81^{+6.78}_{-6.14}$ |
| | FlexLoRA | $13.42^{+15.76}_{-10.37}$ | $87.88^{+0.53}_{-0.84}$ | $70.40^{+1.41}_{-2.02}$ | $74.29^{+1.63}_{-2.54}$ | $90.01^{+0.65}_{-0.41}$ | $55.83^{+0.48}_{-0.60}$ | $65.31^{+3.41}_{-2.80}$ |
| | HETLoRA | $48.09^{+2.64}_{-1.86}$ | $91.74^{+0.34}_{-0.23}$ | $78.59^{+3.76}_{-1.88}$ | $77.53^{+3.20}_{-2.36}$ | $85.30^{+0.07}_{-0.06}$ | $60.04^{+2.05}_{-1.56}$ | $73.55^{+2.01}_{-1.33}$ |
| | **Fed-PLoRA** | $\mathbf{59.38^{+0.43}_{-0.79}}$ | $\mathbf{92.35^{+0.31}_{-0.15}}$ | $\mathbf{86.35^{+0.41}_{-0.57}}$ | $\mathbf{87.25^{+2.71}_{-1.46}}$ | $88.54^{+1.42}_{-1.74}$ | $\mathbf{65.46^{+0.60}_{-0.48}}$ | $\mathbf{79.89^{+0.98}_{-0.87}}$ |

Table 2: Comparison of Fed-PLoRA with baselines on GLUE benchmark.

CareQA Arias-Duart et al. (2025). In the math domain, we use MATH Hendrycks et al. (2021) for training and evaluation. For IID data settings, we evenly split the dataset across clients. For non-IID settings, we apply pathological/Dirichlet data partitioning methods.

We compare Fed-PLoRA against four baselines. FedIT Zhang et al. (2023) represents a classic FFT approach that combines FedAvg with LoRA. Since it is designed for homogeneous LoRA, we only apply it in homogeneous experiments, where it serves as the baseline representing FFT without resource constraints. For heterogeneous LoRA, we consider FLoRA Wang et al. (2024), FlexLoRA Bai et al. (2024), and HETLoRA Cho et al. (2024) (see Section 3.4 for their details). For the heterogeneous setting, clients are (by default) evenly divided into three groups with ranks $(1, r_m, R)$, where the maximum $R$ is set as the optimal rank in homogeneous setting, and the middle rank $r_m \in (1, R)$ is chosen so that the average rank $\lfloor \sum_i r_i / v \rfloor$ is at most $R/2$, reflecting realistic resource-limited clients. The number of clients across tasks ranges from 50 to 200. In each training round, 10% of clients are sampled uniformly at random. The main experimental results are reported as the average with upper and lower deviations over three repeat experiments. Additional experimental configurations are provided in Appendix C.

## 4.1 MAIN EXPERIMENTAL RESULTS

Due to page limitations, we present the main results on Natural Instructions (IID and non-IID) with Llama-1B, the GLUE benchmark (IID) with BERT-base, and financial datasets (IID) with Llama-3.1-8B, while leaving the remaining results on other datasets and non-IID settings in Appendix D.

**Results on Natural Instructions.** Table 1 reports averaged Rouge-L scores of the fine-tuned global model under both IID and non-IID settings (the latter simulated by assigning 20 out of 613 distinct tasks per client). The FedIT results under homogeneous settings show that a larger LoRA rank leads to better fine-tuning performance, improving over the untuned model by +33.06% under the IID setting and by +27.46% under the non-IID setting on average. Fed-PLoRA consistently outperforms heterogeneous baselines. In the IID setting, it achieves average Rouge-L gains of +31.14%, +5.89%, and +6.89% over FLoRA, FlexLoRA, and HETLoRA, respectively, highlighting the effectiveness in eliminating initialization noise and reducing aggregation noise. We also observe that Fed-PLoRA slightly outperforms the homogeneous FedIT with rank $R/2$ by +0.19%. Similar trends appear on other datasets, and in some cases Fed-PLoRA even surpasses FedIT with rank $R$. These effects are often more pronounced under non-IID settings. This suggests that Fed-PLoRA's parallel and randomized module updates ensure that all $R$ global PLoRA modules are updated by different clients across rounds. As a result, rank-wise updates stay aligned across clients, allowing each one-rank module to gradually learn consistent features even under data heterogeneity, consistent with our observations in Figure 2.

| Method | | FPB | FIQA | TFNS | Avg |
|---|---|---|---|---|---|
| Untuned Model | | $50.57^{+0.42}_{-0.31}$ | $29.33^{+0.51}_{-0.44}$ | $49.87^{+0.28}_{-0.36}$ | $40.12^{+0.39}_{-0.27}$ |
| **Homogeneous** | FedIT (Rank=$R$) | $63.53^{+0.33}_{-0.33}$ | $28.94^{+2.30}_{-1.32}$ | $64.11^{+0.45}_{-0.43}$ | $52.19^{+0.48}_{-0.79}$ |
| | FedIT (Rank=$R/2$) | $62.70^{+0.14}_{-0.31}$ | $29.93^{+1.23}_{-2.31}$ | $63.71^{+0.44}_{-0.33}$ | $52.11^{+0.44}_{-0.78}$ |
| **Heterogeneous** $R = 8$ Avg Rank=4 | FLoRA | $52.33^{+0.80}_{-0.68}$ | $31.42^{+0.44}_{-0.24}$ | $50.53^{+0.26}_{-0.45}$ | $44.76^{+0.32}_{-0.28}$ |
| | FlexLoRA | $62.79^{+0.54}_{-0.78}$ | $31.76^{+0.90}_{-0.68}$ | $62.60^{+1.05}_{-1.33}$ | $52.38^{+0.49}_{-0.53}$ |
| | HETLoRA | $60.06^{+0.58}_{-0.33}$ | $\mathbf{32.01^{+2.52}_{-1.42}}$ | $60.09^{+0.58}_{-0.46}$ | $50.72^{+0.92}_{-0.51}$ |
| | **Fed-PLoRA** | $\mathbf{63.94^{+1.15}_{-1.16}}$ | $31.68^{+2.85}_{-2.08}$ | $\mathbf{64.19^{+1.26}_{-0.92}}$ | $\mathbf{53.27^{+1.11}_{-0.85}}$ |

Table 3: Comparison of Fed-PLoRA with baselines on financial datasets.

**Results on GLUE.** As shown in Table 2, our method, Fed-PLoRA, demonstrates substantial improvements over these baselines on IID GLUE benchmark. Fed-PLoRA improves untuned model by +38.62% on average. Compared to FLoRA, Fed-PLoRA achieves average improvements of +63.46% on CoLA and +42.48% on MRPC. This significant margin shows the detrimental impact of FLoRA's random initialization, which incurs significant perturbation to the local training at every round. Compared to FlexLoRA, Fed-PLoRA achieves a notable average improvement of +4.47% on the SST-2 dataset. This suggests that while FlexLoRA aims to provide representative low-rank matrices with SVD, it can still suffer from significant information loss that leads to initialization and aggregation noise, particularly when some clients are constrained to very small LoRA ranks. Fed-PLoRA also outperforms HETLoRA by +9.72% on QQP and +5.42% on RTE. This advantage can be attributed to the large initialization and aggregation noises in HETLoRA from zero-padding and truncation, which Fed-PLoRA is designed to mitigate more effectively. Note that FlexLoRA outperforms FedIT with rank $R$ on QNLI, likely because this relatively simple binary QA task can be well captured using a low rank, making SVD-based aggregation particularly effective.

**Results on Financial Datasets.** As shown in Table 3, Fed-PLoRA achieves average gains of +13.15% over untuned model, +8.51% over FLoRA, +0.89% over FlexLoRA, and +2.55% over HETLoRA. Additionally, both Fed-PLoRA and FlexLoRA here outperforms the homogeneous FedIT baselines. As we discussed before, Fed-PLoRA achieves this by maintaining rank-wise alignment of local module updates across clients. FlexLoRA benefits from its SVD-based initialization, which aligns low-rank modules with the most informative global directions. When the task is inherently low-rank, this targeted representation may outperform homogeneous FedIT, which treats all directions equally.

## 4.2 ADDITIONAL DISCUSSIONS

**Adaptability of the PLoRA to Homogeneous Settings and LoRA Variants.**

Theoretically, PLoRA achieves the same adaptation effect and parameter efficiency as standard multi-rank LoRA (see Section 3.2) and can be extended to other LoRA variants. We adapt it to rsLoRA Kalajdzievski (2023), forming Fed-PrsLoRA, and compare with FedIT and another homogeneous method FFA-LoRA Sun et al. (2024) (which only shares $B$ matrices). We evaluate on IID CoLA dataset, the hardest GLUE task, under homogeneous settings with different ranks. From Figure 3, we observe that Fed-PrsLoRA consistently outperforms Fed-PLoRA, leveraging rsLoRA's improvements over LoRA. Both methods also achieve substantial gains over FFA-

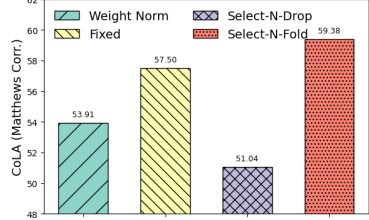

Figure 3: Performance of PLoRA and Fed-PrsLoRA in homogeneous settings.

LoRA (e.g., +10.63% and +10.89% at rank 16) and in some cases surpass FedIT. This advantage becomes more pronounced as the rank decreases, underscoring the effectiveness of PLoRA's parallelization in resource-limited settings.

**Effectiveness of Select-N-Fold.** We also evaluate other strategies for selecting trainable one-rank PLoRA modules in Fed-PLoRA on IID CoLA dataset. We consider: Weight Norm, which picks modules with the largest weight norms; Fixed, which always selects the first $r_i$ modules; and Select-N-Drop, which randomly selects $r_i$ modules like Select-N-Fold but discards the rest. As shown in Figure 4, both random strategies

Figure 4: Select-N-Fold vs. other selection methods.

(Select-N-Drop and Select-N-Fold) outperform the deterministic ones, since randomness ensures that all global PLoRA modules are eventually updated. Select-N-Fold achieves the highest performance by additionally reusing the latest unselected modules during local training.

**Communication and Computational Efficiency.** Figure 5 shows the accuracy of four heterogeneous methods on the IID QQP dataset with respect to communication (uplink and downlink) cost and computation (wall-clock training time) cost. Overall, Fed-PLoRA achieves higher communication efficiency although it incurs additional $R - r_i$ downlink cost compared to others. Both Fed-PLoRA and HETLoRA achieve good computational efficiency as they are lightweight. FlexLoRA incurs substantial computational overhead due to the use of SVD, nearly doubling the training time. A detailed overhead analysis is given in *Appendix Section F.2*.

## 5 RELATED WORK

LoRA-based FFT has recently emerged as a popular paradigm for adapting LLMs while preserving data privacy Zhang et al. (2023); Babakniya et al. (2023); Sun et al. (2024); Zawad et al. (2025). Early approaches such as FedIT Zhang et al. (2023) and FFA-LoRA Sun et al. (2024) combine LoRA with FedAvg or modify update sharing to mitigate aggregation noise, but they assume homogeneous client resources. Other work like FedSA-

LoRA Guo et al. (2024) explores selective aggregation, y
To address resource heterogeneity, several methods allow clients to fine-tune with different LoRA ranks. HET-LoRA Cho et al. (2024) aggregates heterogeneous updates through zero-padding and truncation, FLoRA Wang et al. (2024) stacks full-weight updates before reinitialization, and FlexLoRA Bai et al. (2024) applies SVD-based aggregation with truncation. While effective to some extent, these approaches introduce substantial initialization or aggregation noise, degrading global performance.

The use of multiple parallel LoRA modules has been explored, for enhancing model capacity in centralized training, e.g., Capaboost Haobo et al. (2024) and MELoRA Ren et al. (2024) rather than addressing the unique challenges of federated settings. In contrast, our work designs PLoRA as a mechanism to mitigate initialization and aggregation noise under heterogeneous FFT.

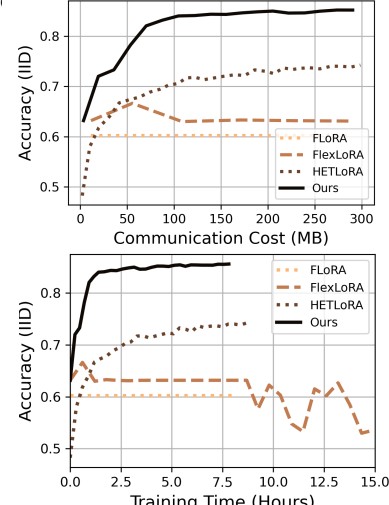

Figure 5: Training efficiency of Fed-PLoRA.

Finally, other efficiency-oriented techniques such as quantization Frantar et al. (2022); Lin et al. (2024), zeroth-order optimization Xu et al. (2024); Zhang et al. (2024a), and system-level methods like activation checkpointing Chen et al. (2016) provide complementary directions for resource-efficient training. These are orthogonal to our design and could be combined with Fed-PLoRA in future work. Due to the limited space, we provide a comprehensive literature review in *Appendix Section B*.

## 6 CONCLUSION

In this paper, we introduced Fed-PLoRA, a novel heterogeneous FFT framework designed to tackle the fundamental challenges of initialization and aggregation noise in LoRA-based fine-tuning. By leveraging PLoRA's parallel one-rank modules together with the Select-N-Fold strategy, Fed-PLoRA aligns client updates more effectively and preserves consistency under resource heterogeneity. We conducted a unified analysis of initialization and aggregation noise of our method, comparing with state-of-the-art heterogeneous LoRA-based FFT methods. Through extensive experiments on multiple tasks, we justified the effectiveness of PLoRA and Select-N-Fold and showed that Fed-PLoRA consistently outperforms existing state-of-the-art methods in both accuracy and efficiency. PLoRA's parallelization design opens new opportunities for integration with other LoRA-like methods, potentially extending its benefits beyond the current framework. In the future, we will work on further reducing the noise that arises during the aggregation of low-rank adaptations and investigating how parallelization can be leveraged to better align model updates under data heterogeneity.

## ACKNOWLEDGMENTS

This work was supported by the National Science Foundation under the Harnessing the Data Revolution for Nevada Fire Science (HDRFS) Seed Grant NSHE-24-37.

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

APPENDIX

This appendix presents additional discussions and experimental details to support the main text. The content is organized as follows:

- Appendix Section A: A code snippet illustrating how our method can be easily integrated into existing frameworks, as well as pseudo-code of Fed-PLoRA.
- Appendix Section B: Extended literature review on related work.
- Appendix Section C: Detailed description of experimental settings.
    - Subsection C.1: Dataset and model descriptions.
    - Subsection C.2: Settings for the main experiments, including FL configuration and infrastructure.
    - Subsection C.3: Settings for the experiments in Figure 7 and Figure 2.
- Appendix Section D: Additional experimental results.
    - Subsection D.1: Results on the GLUE benchmark under the non-IID setting.
    - Subsection D.2: Results on the Dolly-15K dataset under both IID and non-IID settings.
    - Subsection D.3: Results on the Medical Question Answering datasets.
    - Subsection D.4: Results on the MATH reasoning datasets.
    - Subsection D.5: Ablation studies.
        * Subsection D.5.1: Effectiveness of different selection methods for PLoRA modules.
        * Subsection D.5.2: Empirical observations on initialization and aggregation noise.
        * Subsection D.5.3: Impact of the number of clients, heterogeneity ratio, and local rank per client.
        * Subsection D.5.4: Effects of different PLoRA ranks.
        * Subsection D.5.5: Effects of Dropout.
        * Subsection D.5.6: Additional visualizations of cosine similarity across PLoRA modules.
- Appendix Section E: Analysis of initialization and aggregation noise in existing methods.
    - Subsection E.1: Derivations for FLoRA.
    - Subsection E.2: Derivations for FlexLoRA.
    - Subsection E.3: Derivations for HETLoRA.
    - Subsection E.4: Detailed derivations of the aggregation noise for Fed-PLoRA.
- Appendix Section F: Additional Studis.
    - Subsection F.1: Demonstration of the efficiency of the rank-based method on resource-constrained devices.
    - Subsection F.2: Theoretical and numerical analysis of the resource overhead for existing methods and Fed-PLoRA.

USE OF LLM STATEMENT

For the preparation of this manuscript, AI assistants are utilized to aid in checking grammar, spelling, and punctuation.

## A    CODE SNIPPET AND PSEUDO-CODE

```
1  # Implementation
2  # Just one line of code to apply Parallel One-Rank LoRA in PyTorch.
3  model = AutoModelForCausalLM.from_pretrained(model_name)
```

```
4   -  config = LoraConfig(r=8,···)
5   +  config = PLoraConfig(r=1,NumModule=8,···)
6   get_peft_model(model, config)
```

We implement Fed-PLoRA using the Huggingface-style LoRA initialization API (as shown above), making it easy to use by replacing just a single line in the original code.

To better illustrate the proposed method, we present the pseudo-code of Fed-PLoRA as follows.

---

**Algorithm 1** Fed-PLoRA

---

**Require:** Number of communication rounds $T$, global rank $R$, local ranks $\{r_i\}_{i \in [v]}$, pre-trained backbone $\boldsymbol{\Theta}^0$, initial global PLoRA parameters $\boldsymbol{\theta}^0$
1: Server sends $\boldsymbol{\Theta}^0$ to all clients
2: **for** $t = 1$ to $T$ **do**
3:     Server samples a set of clients $\mathcal{S}^t \subseteq [v]$
4:     Server sends global PLoRA parameters $\boldsymbol{\theta}^{t-1}$ to $i \in \mathcal{S}^t$
5:     **for** each client $i \in \mathcal{S}^t$ **in parallel do**
6:         Randomly sample a subset $\mathcal{K}_i^t$ of size $r_i$ from index set $[R]$
7:         Initialize local trainable PLoRA modules: $\boldsymbol{\theta}_i^{t-1} \leftarrow \{\boldsymbol{A}_{(j)}^{t-1}, \boldsymbol{B}_{(j)}^{t-1}\}_{j \in \mathcal{K}_i^t}$
8:         Fold unselected modules into the frozen target weights: $\boldsymbol{\mathcal{W}}_i^t \leftarrow \boldsymbol{\mathcal{W}}^0 + \sum_{j \notin \mathcal{K}_i^t} \boldsymbol{B}_{(j)}^{t-1} \boldsymbol{A}_{(j)}^{t-1}$
             to obtain the local frozen backbone $\boldsymbol{\Theta}_i^t$
9:         Obtain new local PLoRA parameters: $\boldsymbol{\theta}_i^t \leftarrow \mathsf{LocalUpdate}(\boldsymbol{\theta}_i^{t-1}, D_i, \boldsymbol{\Theta}_i^t)$
10:     **end for**
11:     Rank-wise aggregation of PLoRA modules: $\boldsymbol{\theta}^t \leftarrow \mathsf{Agg}(\{\boldsymbol{\theta}_i^t\}_{i \in \mathcal{S}^t}, \{\mathcal{K}_i^t\}_{i \in \mathcal{S}^t})$
12:     Update global model: $\boldsymbol{\Phi}_g^{t+1} \leftarrow \{\boldsymbol{\Theta}^0, \boldsymbol{\theta}^t\}$
13: **end for**
14: **return** $\boldsymbol{\Phi}_g^T = \{\boldsymbol{\Theta}^0, \boldsymbol{\theta}^T\}$

---

## B    COMPREHENSIVE LITERATURE REVIEWS

**Federated LoRA-based Fine-tuning.** The application of FL to LoRA-based fine-tuning of LLMs has become a prominent paradigm, enabling multiple clients to collaboratively adapt models without sharing their raw data Zhang et al. (2023); Babakniya et al. (2023); Sun et al. (2024); Zhang et al. (2024c). Several prior works have investigated this combination: FedIT Zhang et al. (2023) introduces a naive federated fine-tuning method with LoRA and FedAvg, serving as an effective benchmark for further research. FFA-LoRA Sun et al. (2024) introduces a strategy of freezing the LoRA $\mathbf{A}$ matrix during local training to help mitigate aggregation noise at the server. Furthermore, FedSA-LoRA Guo et al. (2024) explores selective aggregation for a personalized FL system, which is designed to only share the $\mathbf{A}$ matrix with the server for aggregation, due to its specific role in learning general knowledge. However, these methods assume homogeneous client resources, which often do not align with the varied resources present in practical FL systems.

**Federated Fine-Tuning with Heterogeneous LoRA Ranks.** Resource heterogeneity remains a significant challenge in federated fine-tuning. To address this, HETLoRA Cho et al. (2024) allows clients to train with different LoRA ranks based on their capacity. For aggregation, local model modules from smaller ranks are zero-padded, and the resulting global LoRA module is subsequently truncated for client LoRA module initialization. However, this process introduces both initialization and aggregation noise due to the limited LoRA rank. FLoRA Wang et al. (2024) employs a stacking-based aggregation method, which is designed to accurately aggregate LoRA module updates from clients with heterogeneous ranks in a full-weight space. While this aims for optimal aggregation, FLoRA then applies this update to the client's target module, and the trainable LoRA modules for the next round are typically initialized from scratch, thereby incurring significant initialization noise. FlexLoRA Bai et al. (2024) utilizes SVD during aggregation, and then initializes LoRA modules for

clients with varying ranks by truncation. These processes lead to both aggregation and initialization noise.

**Parallel Low-Rank Adaptation and Sparse LoRA Fine-Tuning.** The concept of employing multiple parallel LoRA modules has been investigated primarily for enhancing model capacity. For instance, Capaboost Haobo et al. (2024) applies different random masks to a single large rank LoRA module during training, effectively simulating an ensemble of parallel LoRA modules to increase model capacity without incurring additional parameter costs. MELoRA Ren et al. (2024) adopts a group of mini LoRA modules to obtain sparse $\mathbf{A}$ and $\mathbf{B}$ matrices. They claim that each LoRA module is designed to capture different features. These approaches leverage the parallelism of the LoRA module for capacity scaling. In contrast, our framework introduces parallel one-rank LoRA modules specifically as a way to improve initialization and aggregation within the FL system.

Another related line of work Zhou et al. (2021); Zhang et al. (2022b) studies sparse LoRA fine-tuning, where only a subset of LoRA rows is updated while the remaining rows are simply frozen. In principle, such sparsification yields similar training dynamics on the active rows, since the inactive rows receive zero gradients. However, this design has several system-level limitations. First, the frozen rows must still be stored in GPU memory and participate in the forward pass, which increases activation storage and peak memory consumption compared with physically removing these rows. Second, the resulting sparsity pattern is typically unstructured: it does not match the $N : M$ structured sparsity patterns required by modern accelerators (for example, the 2:4 pattern used by NVIDIA A100 sparse tensor cores Zhou et al. (2021)). As a result, simply freezing $R - r_i$ out of $R$ rows cannot exploit hardware sparsity support or lead to real speedups. In contrast, Fed-PLoRA folds unused one-rank modules into the backbone weights, preserving the same effective training behavior on active ranks while avoiding extra computation and memory overhead, which makes it more suitable for large-scale federated deployment.

**Other Resource-Efficient Fine-Tuning Pathways.** Quantization is a widely studied technique for improving inference efficiency by reducing the precision of model weights and activations. Common methods include post-training quantization (PTQ), quantization-aware training (QAT), and activation quantization. PTQ Frantar et al. (2022); Lin et al. (2024); Xiao et al. (2023) applies quantization to a pretrained model without modifying its training process. It is simple and efficient but may lead to noticeable accuracy degradation, especially for low-bit quantization or in sensitive tasks. QAT Nagel et al. (2022) is particularly focused on preserving inference-time accuracy by simulating low-precision operations (e.g., 8-bit or 4-bit) during training through the insertion of fake quantization nodes in the computation graph, allowing the model to adapt to quantization effects. However, QAT typically targets deployment-time efficiency and does not substantially reduce training-time memory or compute costs. Activation quantization Choi et al. (2018) focuses on reducing the bit-width of intermediate activations in addition to weights. While this offers additional memory savings during training, it also introduces additional quantization noise and quantization and de-quantization costs with extra computation. This can make training more unstable and often requires careful calibration to preserve performance.

Zeroth-order optimization Xu et al. (2024) reduces memory usage by avoiding explicit gradient computation. Since it does not require backpropagation, it eliminates the need to store intermediate activations during the forward pass. However, this benefit comes with notable trade-offs: zeroth-order methods typically require many more function evaluations (i.e., higher query complexity) to estimate gradients indirectly, and the resulting gradient estimates tend to be noisier. These factors can significantly slow down convergence, particularly for large, high-dimensional models like LLMs, where accurate and efficient gradient information is crucial for effective training Zhang et al. (2024a).

In addition to algorithmic approaches, several system-level engineering strategies have been developed to address the memory and compute bottlenecks of large model fine-tuning. These methods aim to optimize training pipelines without modifying model architectures or learning objectives. For example, activation checkpointing Chen et al. (2016) reduces GPU memory usage by storing only a subset of activations during the forward pass and recomputing the rest during backpropagation. While it enables training deeper models under tight memory budgets, the trade-off is increased computational overhead due to repeated forward computations. This may lengthen training time significantly, especially in models with expensive forward passes.

All the above techniques are orthogonal and potentially complementary to our approach and could be integrated in specific scenarios. However, in the context of our work, these comparisons are less directly relevant to serve as comparisons. We will consider exploring such combinations in our future work.

## C  EXPERIMENTAL SETTINGS

### C.1  DATASETS AND MODELS

**General Language Understanding Task.** For evaluating general language understanding proficiency, we follow Hao et al. Haobo et al. (2024) that utilize six well-established datasets from the General Language Understanding Evaluation (GLUE) benchmark, as detailed by Wang et al. Wang et al. (2018). Specifically, the tasks included the Corpus of Linguistic Acceptability (CoLA), the Stanford Sentiment Treebank (SST-2), the Microsoft Research Paraphrase Corpus (MRPC), the Quora Question Pairs (QQP) dataset, Question NLI (QNLI), and the Recognizing Textual Entailment (RTE) dataset. Across all these GLUE tasks, the BERT-base model, introduced by Devlin et al. Devlin et al. (2019), is employed as the foundational pre-trained language model.

**General Instruction Following Task.** To assess the framework's performance on tasks requiring general instruction following, we benchmark on two prominent datasets, following the approach of Bai et al. Bai et al. (2024). The first dataset is Natural Instructions (NI), a large-scale collection of diverse NLP tasks structured as instructions, developed by Wang et al. Wang et al. (2022). For this dataset, we utilized a Llama-1B model, with weights sourced from the Data-Juicer project by Chen et al. Chen et al. (2024). The second dataset is Dolly-15K, an open-source dataset of instruction-followed records created by Databricks Conover et al. (2023), for which the OPT-1.3B model from Zhang et al. Zhang et al. (2022a) is employed.

**Domain-Specific Tasks.** We further extend our evaluation to domain-specific applications, focusing on the challenging medical and financial domains. **Medical Domain:** Adhering to the experimental design outlined by Ye et al. Ye et al. (2024) and Flower Gao et al. (2025), we utilize the MedAlpaca dataset, curated by Han et al. Han et al. (2023), for the training phase of our medical domain tasks. The framework's effectiveness is then evaluated on four widely recognized medical question answering benchmarks: PubMedQA by Jin et al. Jin et al. (2019), MedMCQA by Pal et al. Pal et al. (2022), MedQA (USMLE-style questions) also by Jin et al. Jin et al. (2021), and CareQA Arias-Duart et al. (2025). For these demanding medical tasks, we employ the Mistral-7B-v0.3 model MistralAI (2023), leveraging QLoRA for 4-bit precision fine-tuning, as proposed by Dettmers et al. Dettmers et al. (2023). **Financial Domain:** For the financial domain, training is conducted on a financial sentiment analysis dataset derived from the FinGPT project by Yang et al. FinGPT (2023). Evaluation is performed on three established financial NLP benchmarks: the Financial PhraseBank (FPB) by Malo et al. Malo et al. (2014), the Financial Sentiment Analysis on News Headlines (FIQA) dataset by Maia et al. Maia et al. (2018), and the Twitter Financial News Sentiment (TFNS) dataset Neural Magic (2022). These financial experiments are carried out using the Llama-3.1-8B model from Meta Meta AI (2024), also fine-tuned with QLoRA in 4-bit precision.

**Reasoning Tasks.** We additionally evaluate our framework on mathematical reasoning, using the MATH benchmark introduced by Hendrycks et al. Hendrycks et al. (2021). This dataset contains 12,500 competition-style math problems, each tagged by subject (algebra, counting and probability, geometry, number theory, etc.) and difficulty level from 1 to 5. Unlike standard word-problem benchmarks, MATH emphasizes multi-step deduction and symbolic manipulation rather than direct pattern matching, making it a strong stress test for federated parameter-efficient tuning. Every problem includes a full step-by-step solution, enabling training signals beyond final answer supervision. For these math tasks, we employ the Qwen3-4B-Instruct-2507 model, leveraging QLoRA for 4-bit precision fine-tuning, similar to the previous model settings.

**Data Heterogeneity.** To evaluate the performance under diverse data distributions, we specifically assess task heterogeneity using the Natural Instructions, Dolly-15K, and GLUE datasets, as they naturally consist of varied instruction-based or classification-based tasks. We apply both pathological and Dirichlet partitioning methods: the pathological method skews the number of tasks or label categories available to each client, while the Dirichlet method primarily controls the number of samples per client. For Natural Instructions, we use an extreme case where each client is assigned 20

out of 612 tasks; for Dolly-15K, each client receives data from only 1 out of 7 tasks. In the GLUE setup, we apply Dirichlet partitioning with $\alpha = 0.01$, resulting in a highly imbalanced distribution of samples across clients. For all other datasets, we adopt IID partitioning.

## C.2 MAIN RESULT SETTINGS

Table 4: FL settings for all experiments.

| Dataset/Configuration | Model | LoRA | Client | Training Round | Learning Rate | batch size |
|---|---|---|---|---|---|---|
| **CoLA** | BERT-base | rank=1/4/16, lora_alpha=1/4/16, target_modules=["query", "value"] | 100 (10%) | 200 | 0.001 | 16 |
| **SST-2** | BERT-base | rank=1/4/16, lora_alpha=1/4/16, target_modules=["query", "value"] | 100 (10%) | 45 | 0.001 | 16 |
| **MRPC** | BERT-base | rank=1/4/16, lora_alpha=1/4/16, target_modules=["query", "value"] | 100 (10%) | 150 | 0.001 | 16 |
| **QQP** | BERT-base | rank=1/4/16, lora_alpha=1/4/16, target_modules=["query", "value"] | 100 (10%) | 150 | 0.001 | 16 |
| **QNLI** | BERT-base | rank=1/4/16, lora_alpha=1/4/16, target_modules=["query", "value"] | 100 (10%) | 50 | 0.001 | 16 |
| **RTE** | BERT-base | rank=1/4/16, lora_alpha=1/4/16, target_modules=["query", "value"] | 100 (10%) | 150 | 0.001 | 16 |
| **Natural Instruction** | Llama (Data-Juicer-1B) | rank=1/8/16, lora_alpha=1/8/16, target_modules=["q_proj", "v_proj"] | 50 (10%) | 25 | 0.00001 | 4 |
| **Dolly-15K** | OPT-1.3B | rank=1/8/32, lora_alpha=1/8/32, target_modules=["q_proj", "v_proj"] | 50 (10%) | 25 | 0.00001 | 4 |
| **Medical** | Mistral-7B-v0.3 | rank=1/2/8, lora_alpha=1/2/8, target_modules=["q_proj", "v_proj"] | 50 (10%) | 50 | 0.00005 | 4 |
| **Finance** | Llama-3.1-8B | rank=1/2/8, lora_alpha=1/2/8, target_modules=["q_proj", "v_proj"] | 50 (10%) | 15 | 0.00005 | 4 |
| **MATH** | Qwen3-4B-Instruct-2507 | rank=1/2/8, lora_alpha=1/2/8, target_modules=["q_proj", "v_proj"] | 50 (10%) | 15 | 0.00005 | 4 |

**FL Settings.** As shown in Table 4, the FL configurations for all experiments are comprehensively detailed in Table 4. This table outlines the specific models, LoRA parameters, client setups, and training hyperparameters employed for each dataset.

For the GLUE benchmark tasks (CoLA, SST-2, MRPC, QQP, QNLI, and RTE), the BERT-base model is utilized. The heterogeneous ranks are set using $R = 16, r_m = 4$. These experiments involve 100 clients. Instruction fine-tuning tasks on Natural Instruction and Dolly-15K employed Llama (Data-Juicer-1B) and OPT-1.3B models, respectively. For Natural Instruction, $R = 16$ and $r_m = 8$, while for Dolly-15K, $R = 16$ and $r_m = 8$. These setups use 50 clients. Domain-specific experiments on Medical and Finance datasets also involve 50 clients. Medical tasks use Mistral-7B-v0.3, Finance tasks use Llama-3.1-8B, and both have $R = 16$ and $r_m = 2$. For mathematical reasoning, we further evaluate on the MATH dataset using Qwen3-4B-Instruct-2507 under the same federated client setup. All experiments sample 10% of clients uniformly at random per round.

Across these diverse setups, the number of training rounds was task-specific, varying from 15 to 200, as detailed in Table 4. Learning rates are selected for each task group after evaluating a common search space of $\{0.005, 0.001, 0.0005, 0.0001, 0.00005, 0.00001\}$; specifically, GLUE tasks utilize a learning rate of 0.001, Natural Instruction and Dolly-15K tasks are trained with 0.00001, and both Medical, Finance, and MATH experiments employ a learning rate of 0.00005. Batch sizes are also tailored: 16 for GLUE tasks, 4 for instruction fine-tuning, and 4 for the domain-specific experiments.

*Justification heterogeneous rank settings:* We empirically evaluate performance across a wide range of LoRA ranks $\{1, 2, 4, 8, 16, 32, 64\}$. For the highest-resource clients, we select the largest rank that does not lead to overparameterization, following standard practice in FL to ensure a balance between model capacity and resource capacity. For clients with mid-level resources, we adopt middle ranks so that the average rank does not exceed the half of the highest rank, reflecting that majoraty of the clients have low resources.

*Justification for rank settings:* In homogeneous settings, we evaluate LoRA ranks $\{1, 2, 4, 8, 16, 32, 64\}$ to identify the optimal rank. In heterogeneous settings, high-resource clients are assigned this optimal rank to avoid over-parameterization while maintaining a balance between model capacity and device constraints. Low-resource clients are fixed at rank 1. Mid-resource clients are assigned intermediate ranks such that the average rank does not exceed half of the maximum, reflecting the realistic case where most clients are resource-limited.

We evaluate performance across a broad range of LoRA ranks $\{1, 2, 4, 8, 16, 32, 64\}$ in homogeneous settings and identify the best rank. For high-resource clients in heterogeneous settings, we assign the best rank that avoids over-parameterization, ensuring a balance between model capacity and device constraints. For mid-resource clients, we choose intermediate ranks such that the average rank does not exceed half of the maximum rank, reflecting the realistic scenario where most clients have limited resources.

**Implementation Details.** Our framework was implemented using PyTorch version 2.6.0 (built with CUDA 12.4 support). Key libraries included Hugging Face `transformers` version 4.51.3,

`datasets` version 3.6.0, and `peft` version 0.9.0 for LoRA and QLoRA functionalities. Experiments were conducted using CUDA 12.4. All experiments were carried out on a server equipped with an AMD EPYC 7763 64-Core Processor, 1.0 TB of system RAM, and 8 × NVIDIA RTX A6000 GPUs. The total GPU hours for running all the experiments are over 5,000.

### C.3 OTHER EXPERIMENTAL SETTINGS

The experiments illustrated in Figure 7 are conducted on the QQP, MRPC, and RTE datasets using a BERT-based model. A key aspect of this setup is the use of a homogeneous LoRA rank across all clients (i.e., every client uses the same rank). Two distinct homogeneous rank configurations are evaluated: a 'Large Rank' setup with a uniform rank of 16 per client, and a 'Small Rank' setup with a uniform rank of 1 per client. These FL experiments involved a total of 100 clients, with 10% of clients participating in each training round. The training is conducted for a total of 150 rounds. In Figure 2, we report the average cosine similarities calculated for LoRA **A** matrices on the Query target module of Layer 1. These calculations are based on experiments performed on the RTE dataset, with a non-IID data distribution achieved by assigning data from only one category to each client.

## D ADDITIONAL EXPERIMENTAL RESULTS

### D.1 RESULTS ON NON-IID GLUE DATASET

Table 5: Evaluation results on GLUE datasets under a non-IID setting (Dirichlet $\alpha = 0.01$).

| Method | | CoLA Matthews Corr. | SST-2 Acc. | MRPC Acc. | QQP Acc. | QNLI Acc. | RTE Acc. | Avg |
|---|---|---|---|---|---|---|---|---|
| Untuned Model | | 1.20 | 49.08 | 31.61 | 63.09 | 49.95 | 52.70 | 41.27 |
| **Homogeneous** | FedIT (Rank=$R$) | 56.27 | 90.25 | 85.29 | 87.07 | 87.07 | 64.62 | 78.28 |
| | FedIT (Rank=$R/2$) | 51.80 | 90.25 | 79.31 | 84.23 | 88.01 | 61.42 | 75.67 |
| **Heterogeneous** $R = 16$ Avg Rank=7 | FLoRA | −2.07 | 88.97 | 68.38 | 38.75 | 47.37 | 47.29 | 48.18 |
| | FlexLoRA | 38.51 | 90.25 | 68.39 | 79.91 | 70.98 | 53.79 | 66.97 |
| | HETLoRA | 40.08 | 89.97 | 76.35 | 81.71 | 85.96 | 58.12 | 72.03 |
| | **Fed-PLoRA** | **52.33** | **90.71** | **79.65** | **84.79** | **88.01** | **61.01** | **76.08** |

Table 5 presents evaluation results on the GLUE dataset under a non-IID setting, where client data distributions are skewed using a Dirichlet distribution with $\alpha = 0.01$, representing a highly imbalanced and extreme case. In this challenging scenario, our method Fed-PLoRA consistently outperforms all baselines. Compared to FLoRA, Fed-PLoRA yields substantial gains of +54.40% on CoLA and +11.27% on MRPC, underscoring the limitations of FLoRA's random initialization and emphasizing the benefit of a more carefully designed initialization strategy in our method. Compared to FlexLoRA, Fed-PLoRA achieves a notable improvement of +17.03% on QNLI and +0.46% on SST-2. Although FlexLoRA leverages SVD to derive representative low-rank matrices, it remains susceptible to information loss, which can lead to significant initialization and aggregation noise, especially when some clients operate with much lower ranks than the global configuration and under non-IID cases. When compared to HETLoRA, Fed-PLoRA improves performance by +3.08% on QQP and +2.89% on RTE. These gains come from Fed-PLoRA's more effective handling of initialization and aggregation noise, which HETLoRA introduces through zero-padding and rank truncation. Furthermore, Fed-PLoRA surpasses FedIT with a fixed rank of 8 by an average of +0.41%, despite using a lower average rank of 7. This demonstrates the efficiency and adaptability of our approach that incorporate all resource-constrained clients into training with zero initialization noise and low aggregation noise.

### D.2 RESULTS ON DOLLY-15K DATASET

Table 6 presents the main results on the Dolly-15K dataset. We report Rouge-L scores of the fine-tuned global model under both IID and non-IID settings, where the non-IID scenario is simulated

Table 6: Evaluation results on IID and non-IID Dolly-15K datasets with OPT-1.3B model.

| Method | | Dolly-15K (Rouge-L) | |
|---|---|---|---|
| | | IID | non-IID |
| Untuned Model | | 40.02 | |
| **Homogeneous** | FedIT (Rank=$R$) | 60.05 | 59.37 |
| | FedIT (Rank=$R/2$) | 59.75 | 59.07 |
| **Heterogeneous** $R = 32$ Avg Rank=13 | FLoRA | 40.01 | 40.01 |
| | FlexLoRA | 57.40 | 48.82 |
| | HETLoRA | 59.38 | 58.36 |
| | **Fed-PLoRA** | **60.07** | **59.69** |

by assigning each client data from only 1 out of 7 distinct tasks. Our method, Fed-PLoRA, consistently outperforms all baseline approaches under both data and resource heterogeneity. Notably, Fed-PLoRA surpasses FlexLoRA in the non-IID setting by a large margin of +10.87%, highlighting the critical impact of initialization noise in federated fine-tuning. Moreover, Fed-PLoRA even outperforms the homogeneous case with a full rank 32, achieving gains of +0.02% and +0.32% under IID and non-IID conditions, respectively. This suggests that our parallel one-rank LoRA modules offer greater flexibility than conventional high-rank LoRA, and can potentially lead to better overall performance. These results further demonstrate the effectiveness of Fed-PLoRA in mitigating initialization and aggregation noise via its parallelized design.

## D.3 RESULTS ON MEDICAL DATASETS

Table 7: Evaluation results on medical datasets with Mistral-7B-v0.3 model.

| Method | | PubMedQA | MedMcQA | MedQA | CareQA | Avg |
|---|---|---|---|---|---|---|
| Untuned Model | | 55.77 | 38.43 | 40.92 | 41.28 | 44.10 |
| **Homogeneous** | FedIT (Rank=$R$) | 70.60 | 41.16 | 46.19 | 47.35 | 51.32 |
| | FedIT (Rank=$R/2$) | 68.60 | 41.19 | 46.19 | 46.55 | 50.63 |
| **Heterogeneous** $R = 8$ Avg Rank=3 | FLoRA | 55.80 | 40.30 | 45.70 | 46.10 | 46.98 |
| | FlexLoRA | 58.40 | 41.16 | **46.26** | 46.26 | 48.02 |
| | HETLoRA | 69.80 | 41.23 | **46.26** | 46.10 | 50.85 |
| | **Fed-PLoRA** | **70.20** | **41.50** | 46.11 | **46.85** | **51.16** |

Table 7 presents the main results (in accuracy) on medical and financial datasets. It is noteworthy that due to the inherent capabilities of the LLMs employed and the potential domain shift between the training and test sets, the observed performance gaps among different federated methods remain relatively small. Fed-PLoRA demonstrates improvements over FLoRA, achieving accuracy gains of +14.40% on PubMedQA, +1.20% on MedMcQA, +0.75% on CareQA, +12.29% on FPB, and +13.15% on TFNS. However, when compared to FlexLoRA and HETLoRA on the MedQA dataset and to FLoRA on the FIQA dataset, our method shows slight accuracy drops. This may be attributed to the nature of these domain-specific instruction fine-tuning tasks with large LLMs requiring only a small LoRA rank. In such scenarios, the initialization noise in FlexLoRA and HETLoRA can be relatively small.

## D.4 RESULTS ON MATH DATASETS

Table 8 summarizes the performance of various federated fine-tuning methods across seven MATH sub-categories. The baseline (untuned model) exhibits limited reasoning capability across most categories, with overall average accuracy remaining below 10%. Under homogeneous settings, FedIT with full rank ($R$) yields consistent gains over the $R/2$ configuration across all sub-tasks, demonstrating the benefit of maintaining richer update capacity during global aggregation.

Table 8: Evaluation of Qwen3-4B-A3B-Instruct-2507 on MATH datasets.

| Method | | Algebra | Counting&Prob. | Geometry | Inter. Algebra | Number Theory | PreAlgebra | PreCalculus | Avg |
|---|---|---|---|---|---|---|---|---|---|
| **Untuned** | | 12.38 | 5.90 | 3.75 | 1.32 | 5.74 | 15.61 | 4.57 | 7.94 |
| **Homogeneous** | FedIT (Rank=$R$) | 42.20 | 26.79 | 19.20 | 13.73 | 26.11 | 42.93 | 10.98 | 28.38 |
| | FedIT (Rank=$R/2$) | 36.47 | 22.57 | 16.70 | 11.62 | 21.48 | 37.08 | 12.27 | 24.62 |
| **Heterogeneous** $R = 8$ Avg Rank=4 | FLoRA | 12.72 | 6.75 | 4.38 | 1.88 | 5.37 | 15.04 | 2.93 | 7.94 |
| | FlexLoRA | 38.83 | 19.62 | 15.65 | 9.52 | 22.03 | 40.41 | 10.62 | 24.88 |
| | HETLoRA | 28.47 | 12.23 | 11.69 | 9.85 | 17.77 | 24.79 | 10.07 | 18.16 |
| | **Fed-PLoRA** | **43.80** | **24.89** | **18.99** | **14.17** | **27.77** | **42.59** | **12.82** | **28.96** |

Under heterogeneous environments where the average rank is fixed to $R = 4$, Fed-PLoRA achieves the strongest overall results, outperforming FLoRA, FlexLoRA, and HETLoRA with significant improvements in Algebra, Counting & Probability, and Number Theory, where gains exceed +5–15% on average. Notably, Fed-PLoRA also maintains competitive accuracy on Geometry and Intermediate Algebra, indicating enhanced adaptability even under rank-diverse client configurations.

## D.5 ABLATION STUDIES

### D.5.1 COMPARING SELECT-N-FOLD WITH OTHER STRATEGIES

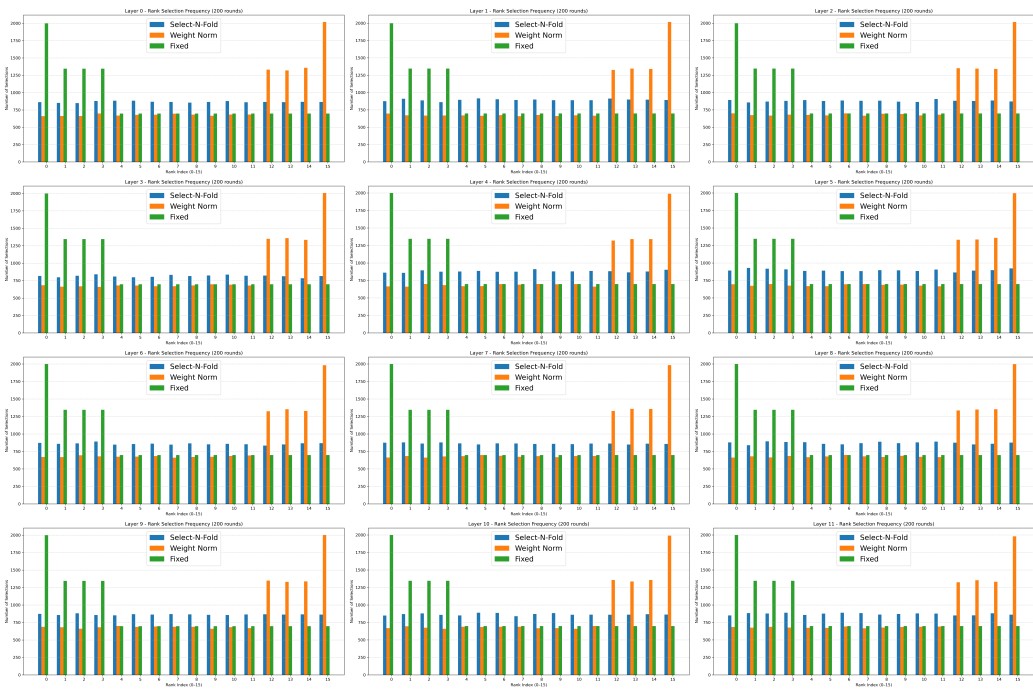

Figure 6: Visualization of rank selection results using different selection strategies.

Here, we provide a visualization of the rank selection results in Figure 6, comparing Select-N-Fold with the Weight Norm and Fixed methods in Figure 4. Overall, random strategies yield a more uniform and unbiased distribution across layers.

### D.5.2 EMPIRICAL OBSERVATIONS ON INITIALIZATION AND AGGREGATION NOISE IN SOTA METHODS

As discussed in Section 3.4, existing federated LoRA-based fine-tuning methods suffer from different levels of initialization and aggregation noise. To isolate these effects, Figure 7 illustrates optimization trajectories and empirical results. In this setup, we fix the LoRA rank across clients to remove initialization noise for FlexLoRA and HETLoRA, allowing a clearer focus on their aggregation noise (detailed settings in *Appendix Section C.3*).

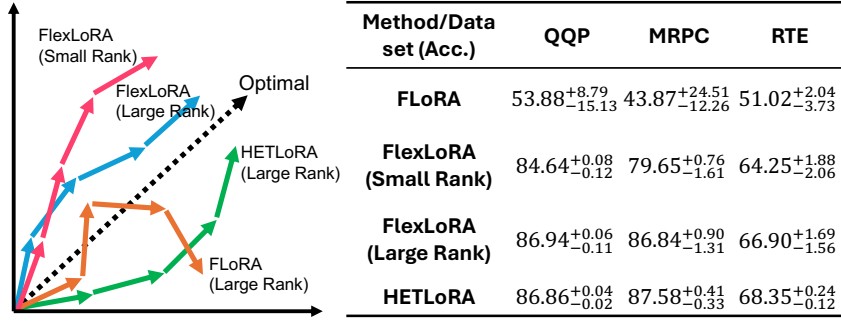

| Method/Data set (Acc.) | QQP | MRPC | RTE |
|---|---|---|---|
| FLoRA | $53.88^{+8.79}_{-15.13}$ | $43.87^{+24.51}_{-12.26}$ | $51.02^{+2.04}_{-3.73}$ |
| FlexLoRA (Small Rank) | $84.64^{+0.08}_{-0.12}$ | $79.65^{+0.76}_{-1.61}$ | $64.25^{+1.88}_{-2.06}$ |
| FlexLoRA (Large Rank) | $86.94^{+0.06}_{-0.11}$ | $86.84^{+0.90}_{-1.31}$ | $66.90^{+1.69}_{-1.56}$ |
| HETLoRA | $86.86^{+0.04}_{-0.02}$ | $87.58^{+0.41}_{-0.33}$ | $68.35^{+0.24}_{-0.12}$ |

Figure 7: The intuitive convergence trajectories of existing methods and empirical results on three datasets in homogeneous settings.

The results show that FLoRA, while free of aggregation noise, struggles due to random initialization, yielding low accuracies of 53.88%, 43.87%, and 51.02% on QQP, MRPC, and RTE. FlexLoRA, by contrast, is mainly affected by its rank-dependent SVD-based aggregation noise: at large ranks (e.g., $r_i = 16$) it performs well (86.94% on QQP), but its accuracy drops at smaller ranks (e.g., 84.64% at $r_i = 1$). HETLoRA introduces bias through its averaging rule, yet achieves strong performance, with 87.58% on MRPC and 68.35% on RTE, suggesting its noise can be no worse than FlexLoRA's. Finally, FlexLoRA incurs heavy computation from repeated SVD, whereas HETLoRA uses a lightweight averaging scheme.

### D.5.3 IMPACT OF HYPERPARAMETERS

Table 9: The impacts of the number of TC and clients' HR.

| Method | CoLA Dataset | | |
|---|---|---|---|
| | TC=100 HR=1:1:1 | TC=200 HR=1:1:1 | TC=100 HR=6:3:1 |
| FLoRA | −4.08 | −2.07 | −2.07 |
| FlexLoRA | 13.42 | −4.73 | −2.50 |
| HETLoRA | 48.09 | 50.54 | 51.04 |
| **Fed-PLoRA** | **59.38** | **55.56** | **58.28** |

**The Impact of Total Clients and Resource Heterogeneity.** We study how the total number of clients (TC) and the heterogeneity ratio (HR) of client resources affect performance. Table 9 reports results on CoLA. Across all configurations, Fed-PLoRA consistently outperforms baselines (FLoRA, FlexLoRA, HETLoRA). Increasing TC from 100 to 200 (with balanced HR=1:1:1) slightly reduces Fed-PLoRA's score (59.38% to 55.56%) but still keeps a clear margin over HETLoRA (50.54%). When shifting HR from balanced (1:1:1) to skewed (6:3:1, more low-resource clients) with TC=100, Fed-PLoRA remains robust (59.38% to 58.28%). These results show that Fed-PLoRA scales well with both client population and resource imbalance.

Table 10: The impacts of the heterogeneous rank setting.

| Method | CoLA Dataset | | | |
|---|---|---|---|---|
| | Ranks 1/2/4 | Ranks 1/2/16 | Ranks 1/4/16 | Ranks 2/8/16 |
| FLoRA | −2.07 | −2.07 | −4.08 | −2.07 |
| FlexLoRA | 1.01 | 4.16 | 13.42 | −3.76 |
| HETLoRA | 47.44 | 46.70 | 48.09 | 49.54 |
| **Fed-PLoRA** | **58.58** | **60.26** | **59.38** | **60.41** |

**The Impact of Client Rank Settings.** We evaluate how different distributions of LoRA ranks across clients influence performance, with results on CoLA shown in Table 10. Rank configurations tested include {1, 2, 4}, {1, 2, 16}, {1, 4, 16}, and {2, 8, 16}, under the same settings as the main experiments. Fed-PLoRA consistently achieves the highest Matthews Correlation scores, outperforming FLoRA, FlexLoRA, and HETLoRA across all cases. For example, Fed-PLoRA attains 58.58% with ranks 1/2/4 and 60.41% with ranks 2/8/16, demonstrating strong robustness and adaptability to heterogeneous client resources.

Table 11: Performance and resource usage of different methods under extremely large global model rank.

| Method | | CoLA Matthews Corr. | SST-2 Accuracy | MRPC Accuracy | RTE Accuracy | Client FLOPs GFLOPS | Server FLOPs MFLOPS | Throughput seconds / 100 tokens | Uplink/Downlink MB / Round |
|---|---|---|---|---|---|---|---|---|---|
| **Homogeneous** | **FedIT (Rank=$R$)** | 62.94 | 91.97 | 88.72 | 73.28 | 695.50 | 5.89 | 2.33 | 430.08 / 430.08 |
| **Heterogeneous** | **FLoRA** | $-4.98$ | 70.52 | 31.51 | 52.70 | 202.40 | 74.94 | 0.89 | 23.52 / 235.20 |
| $R = 128$ | **FlexLoRA** | 1.43 | 50.91 | 68.38 | 53.42 | 202.37 | 620.20 | 0.89 | 23.52 / 23.52 |
| $\max(r_i) = 16$ | **HETLoRA** | 47.65 | 71.62 | 80.88 | 59.56 | 202.37 | 1.71 | 0.89 | 23.52 / 23.52 |
| Avg Rank =7 | **Fed-PLoRA** | **55.43** | **72.82** | **85.04** | **70.36** | 202.37 | 1.71 | 0.89 | 23.52 / 129.36 |

**The Impact of Extremely Large Global Rank $R$ on Learning Efficiency.** We evaluate the effect of increasing the global model rank to 128 while capping local ranks at 16 to reflect realistic resource constraints. Experiments are conducted on four GLUE tasks under the IID setting with a batch size of 16. Input sequence length is set to 128 for CoLA and SST-2, and 256 for MRPC and RTE.

Client-side FLOPS (including initialization and LoRA operations) are averaged over all clients and rounds on a per-sample basis. Server-side FLOPS measure aggregation costs. Throughput is reported as the average seconds to process 100 training tokens on clients. Communication volume denotes uplink and downlink per round per client in MB under 32-bit precision.

As shown in Table 11, Fed-PLoRA consistently outperforms other heterogeneous baselines under this extreme $R$. Client-side FLOPS and throughput remain identical across methods since all use the same model structure and simulated hardware. On the server side, Fed-PLoRA matches HETLoRA's efficiency by aggregating one-rank modules independently. Uplink costs are equally low across all methods, as only trainable LoRA modules are transmitted. The main trade-off is in downlink cost, which scales with $R$ and equals that of FedIT. However, excessively large $R$ is rarely practical under heterogeneous constraints and may even cause divergence, as observed for FLoRA and FlexLoRA on CoLA. Since LoRA is designed for parameter efficiency, setting $R$ close to the hidden dimension provides little benefit and often harms stability.

### D.5.4 THE IMPACT OF DIFFERENT CONFIGURATIONS OF PLORA

We evaluate how the rank size of individual PLoRA modules influences performance. Specifically, we compare our default one-rank design (Fed-PLoRA (Parallel One-Rank)) with a two-rank variant (Fed-PLoRA (Parallel Two-Rank)), both configured to yield the same total effective LoRA rank across clients with heterogeneous rank settings (2, 8, 16). The one-rank configuration achieves a slightly higher Matthews Correlation score (60.41%) than the two-rank configuration (59.35%). This result supports our design choice.

### D.5.5 THE IMPACTS OF DROPOUT

Table 12: The impacts of dropout in the PLoRA.

| Method | Dolly-15K (Rouge-L) Rank: 1/8/32 | |
|---|---|---|
| | IID | non-IID |
| Fed-PLoRA (Unfold, w/ Dropout) | 60.07 | 59.70 |
| **Fed-PLoRA** | 60.07 | 59.69 |

We investigate the impact of applying LoRA dropout within the Fed-PLoRA framework. Typically, dropout layers are utilized for regularization before input features pass into LoRA modules

(specifically, before the LoRA **A** matrix). In our Fed-PLoRA approach, the *Select-N-Fold* strategy involves merging non-selected parallel LoRA modules into the main target module. When these modules are folded, any dropout layers specifically associated with these folded LoRA paths are effectively bypassed for those components in subsequent forward passes. This characteristic motivates an evaluation to understand how performance is affected by dropout under our folding mechanism. Table 12 presents a comparison on the Dolly-15K dataset (Rouge-L score), with client ranks set to (1, 8, 32). We compare a variant termed "Fed-PLoRA (Unfold, w/ Dropout)", where we assume modules are frozen but kept unfolded, allowing standard LoRA dropout to be active on all PLoRA modules against our standard Fed-PLoRA configuration, but with higher computational costs. The results show minimal differences between the two approaches across both IID and non-IID data distributions. In the IID setting, both configurations achieve a Rouge-L score of 60.07%. In the non-IID setting, "Fed-PLoRA (Unfold, w/ Dropout)" achieves 59.70%, while our method achieves 59.69%. These nearly identical results suggest that not using dropout on frozen layers does not affect the performance.

### D.5.6 MORE VISUALIZATION ON COSINE SIMILARITIES ACROSS PLORA MODULES.

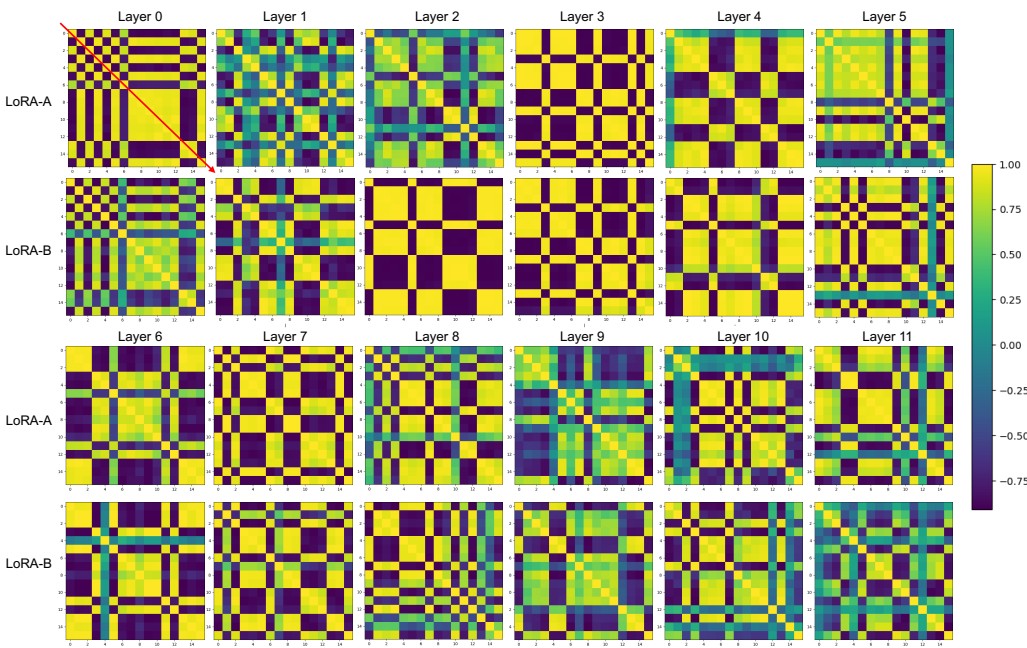

Figure 8: Cosine similarity heatmaps of PLoRA modules on the non-IID RTE dataset (averaged over all rounds). Each block reports pairwise similarities of modules across clients for LoRA-A and LoRA-B, spanning layers 0 through 11.

To analyze the aggregation noise in Fed-PLoRA, we visualize the cosine similarity heatmaps across clients' PLoRA modules for the Query weights in all 12 transformer layers of BERT-base, averaged over all rounds. Both $A$ and $B$ module similarities are shown. According to Section E, when $\mathbf{A}_{i,(j)}^t \approx \overline{\mathbf{A}}_{(j)}^t$ or $\mathbf{B}_{i,(j)}^t \approx \overline{\mathbf{B}}_{(j)}^t$, aggregation noise of Fed-PLoRA is minimized. Figure 8 reveals consistently high similarity values (near 1.0) along diagonal and cross-wise patterns, particularly highlighted by the red arrow in Layer 0's LoRA-A, suggesting strong similarity among certain ranks. Moreover, we observe that some consecutive layers maintain high inter-client similarity, indicating that even under non-IID settings, certain ranks learn overlapping or redundant features, potentially pointing to shared semantic structures across clients.

This pattern emerges naturally from the design of Fed-PLoRA. Since each client receives the full set of $R$ global PLoRA modules and selects a subset for local training, all modules remain aligned in initialization across rounds. The random selection mechanism ensures that over time, every rank-$j$ module is trained by a diverse subset of clients, leading to gradual synchronization of parameters across the population. Furthermore, the frozen folded modules help preserve the global

structure even on clients with limited capacity, reducing noise accumulation across rounds. As a result, frequently selected ranks tend to exhibit strong inter-client alignment, while occasionally selected modules still retain structural consistency due to their shared initialization and partial update history. This design promotes implicit coordination among clients and contributes to the observed high cosine similarity, thereby reducing the aggregation noise without explicit regularization or coordination.

While our method does not entirely eliminate aggregation noise. This is a characteristic shared with several other federated LoRA-based fine-tuning methods, such as FedIT and HETLoRA. However, we note that there is a growing body of research in the literature Sun et al. (2024); Guo et al. (2024) focused on improving the aggregation of LoRA in the homogeneous setting. We will explore the integration of these methods with Fed-PLoRA in heterogeneous settings in our future research.

## E    DETAILED ANALYSIS OF INITIALIZATION AND AGGREGATION NOISE

This section investigates the noise arising from initialization and aggregation processes in heterogeneous federated LoRA-based fine-tuning methods.

We use the following general definitions for noise as established in the main paper: Let $\boldsymbol{\theta}^{t-1} = (\mathbf{A}^{t-1}, \mathbf{B}^{t-1})$ be the global LoRA parameters from the server at the end of round $t-1$ (with global rank $R$). Let $\boldsymbol{\theta}_i^{t-1} = (\mathbf{A}_i^{t-1}, \mathbf{B}_i^{t-1})$ be the local LoRA parameters with which client $i$ (from a set of $v$ clients) starts round $t$ (with local rank $r_i$). The **Initialization Noise** at the beginning of round $t$ is:

$$\mathcal{N}_{\text{Init}}^t := \sum_{i=1}^{v} \left( \left\| \mathbf{A}^{t-1} \ominus \mathbf{A}_i^{t-1} \right\|_F + \left\| \mathbf{B}^{t-1} \ominus \mathbf{B}_i^{t-1} \right\|_F \right)$$

Here, $\mathbf{X} \ominus \mathbf{Y}$ signifies the part of $\mathbf{X}$ not captured by $\mathbf{Y}$.

After local training, client $i$ produces updated LoRA parameters $\boldsymbol{\theta}_i^t = (\mathbf{A}_i^t, \mathbf{B}_i^t)$. The ideal aggregated target module update is $\Delta \mathcal{W}_*^t = \frac{1}{v} \sum_{i=1}^{v} \mathbf{B}_i^t \mathbf{A}_i^t$. The actual aggregated target module update by a specific method is $\Delta \mathcal{W}^t = \mathbf{B}^t \mathbf{A}^t$, where $\mathbf{A}^t, \mathbf{B}^t$ are the global LoRA parameters after aggregation at round $t$. The **Aggregation Noise** at the end of round $t$ is:

$$\mathcal{N}_{\text{Agg}}^t := \left\| \Delta \mathcal{W}_*^t - \Delta \mathcal{W}^t \right\|_F$$

### E.1    FLoRA

FLoRA Wang et al. (2024) employs a stacking-based aggregation. For initialization, client LoRA modules $\mathbf{A}_i^{t-1}$ are randomly initialized (e.g., Gaussian Distribution), and $\mathbf{B}_i^{t-1}$ is initialized to zeros at each round $t$.

**Initialization Noise ($\mathcal{N}_{\text{FLoRA\_Init}}^t$):** Given $\mathbf{A}_i^{t-1}$ is random and $\mathbf{B}_i^{t-1} = \mathbf{0}$, these local parameters do not retain information from the previous global LoRA parameters $\mathbf{A}^{t-1}$ and $\mathbf{B}^{t-1}$. For $\mathbf{B}$ matrices: Since $\mathbf{B}_i^{t-1} = \mathbf{0}$, the part of $\mathbf{B}^{t-1}$ not captured by $\mathbf{B}_i^{t-1}$ is $\mathbf{B}^{t-1}$ itself. Thus, $\left\| \mathbf{B}^{t-1} \ominus \mathbf{B}_i^{t-1} \right\|_F = \left\| \mathbf{B}^{t-1} \right\|_F$. For $\mathbf{A}$ matrices: $\mathbf{A}_i^{t-1}$ is random and independent of $\mathbf{A}^{t-1}$. It does not systematically capture any part of $\mathbf{A}^{t-1}$. The initialization noise is formulated as:

$$\mathcal{N}_{\text{FLoRA\_Init}}^t = \sum_{i=1}^{v} \left( \left\| \mathbf{A}^{t-1} \ominus \mathbf{A}_i^{t-1} \right\|_F + \left\| \mathbf{B}^{t-1} \ominus \mathbf{B}_i^{t-1} \right\|_F \right)$$

$$\approx \sum_{i=1}^{v} \left( \left\| \mathbf{A}^{t-1} \right\|_F + \left\| \mathbf{B}^{t-1} \right\|_F \right) + \sigma^2$$

where $\sigma^2$ denotes the magnitude of the random noise used for initializing $\mathbf{A}$ matrix. This noise is substantial.

**Aggregation Noise ($\mathcal{N}_{\text{FLoRA\_Agg}}^t$):** FLoRA stacks client LoRA modules. Let client $i$ have $\mathbf{A}_i^t \in \mathbb{R}^{r_i \times k}$ and $\mathbf{B}_i^t \in \mathbb{R}^{d \times r_i}$. The server forms:

$$\mathbf{A}^t = \begin{pmatrix} \mathbf{A}_1^t \\ \vdots \\ \mathbf{A}_v^t \end{pmatrix} \in \mathbb{R}^{(\sum r_i) \times k} \quad \text{and} \quad \mathbf{B}^t = \begin{pmatrix} \mathbf{B}_1^t & \cdots & \mathbf{B}_v^t \end{pmatrix} \in \mathbb{R}^{d \times (\sum r_i)}$$

The aggregated update is $\Delta \mathcal{W}_{\text{FLoRA}}^t = \frac{1}{v} \mathbf{B}^t \mathbf{A}^t$. The product is $\mathbf{B}^t \mathbf{A}^t = \sum_{i \in [v]} \mathbf{B}_i^t \mathbf{A}_i^t$. So, the actual aggregated update is:

$$\Delta \mathcal{W}_{\text{FLoRA}}^t = \frac{1}{v} \sum_{i \in [v]} \mathbf{B}_i^t \mathbf{A}_i^t$$

The ideal target module update is $\Delta \mathcal{W}_*^t = \frac{1}{v} \sum_{i \in [v]} \mathbf{B}_i^t \mathbf{A}_i^t$. Therefore, the aggregation noise is:

$$\mathcal{N}_{\text{FLoRA\_Agg}}^t = \left\| \Delta \mathcal{W}_*^t - \Delta \mathcal{W}_{\text{FLoRA}}^t \right\|_F$$
$$= \left\| \frac{1}{v} \sum_{i \in [v]} \mathbf{B}_i^t \mathbf{A}_i^t - \frac{1}{v} \sum_{i \in [v]} \mathbf{B}_i^t \mathbf{A}_i^t \right\|_F = 0$$

### E.2 FLEXLoRA

FlexLoRA aggregates LoRA modules in full weight space, then uses SVD for local LoRA modules. Clients truncate these for initialization.

**Initialization Noise ($\mathcal{N}_{\text{FlexLoRA\_Init}}^t$):** The server has global LoRA modules $\mathbf{A}^{t-1}$ (from SVD, rank $R$) and $\mathbf{B}^{t-1}$ (from SVD, rank $R$). Client $i$ (rank $r_i \leq R$) initializes by truncation: $\mathbf{A}_i^{t-1} = \mathbf{A}_{[:r_i,:]}^{t-1}$ (first $r_i$ rows of $\mathbf{A}^{t-1}$). $\mathbf{B}_i^{t-1} = \mathbf{B}_{[:,:r_i]}^{t-1}$ (first $r_i$ columns of $\mathbf{B}^{t-1}$). The part of $\mathbf{A}^{t-1}$ not captured is $\mathbf{A}_{[r_i+1:R,:]}^{t-1}$. The part of $\mathbf{B}^{t-1}$ not captured is $\mathbf{B}_{[:,r_i+1:R]}^{t-1}$. The initialization noise is then formulated as,

$$\mathcal{N}_{\text{FlexLoRA\_Init}}^t = \sum_{i \in [v]} \left( \left\| \mathbf{A}^{t-1} \ominus \mathbf{A}_i^{t-1} \right\|_F + \left\| \mathbf{B}^{t-1} \ominus \mathbf{B}_i^{t-1} \right\|_F \right)$$
$$= \sum_{i \in [v]} \left( \left\| \mathbf{A}_{[r_i+1:R,:]}^{t-1} \right\|_F + \left\| \mathbf{B}_{[:,r_i+1:R]}^{t-1} \right\|_F \right)$$

This noise is non-zero if any $r_i < R$.

**Aggregation Noise ($\mathcal{N}_{\text{FlexLoRA\_Agg}}^t$):** Clients send $\mathbf{A}_i^t, \mathbf{B}_i^t$. The server computes the ideal average update:

$$\Delta \mathcal{W}_*^t = \frac{1}{v} \sum_{i=1}^{v} \mathbf{B}_i^t \mathbf{A}_i^t$$

However, FlexLoRA then performs SVD on $\Delta \mathcal{W}_*^t$ to get a rank-$R$ approximation:

$$\Delta \mathcal{W}_{\text{FlexLoRA}}^t = \text{SVD}(\Delta \mathcal{W}_*^t) = \mathbf{U}^t \mathbf{S}^t (\mathbf{V}^t)^\top$$

The aggregation noise is the SVD truncation error:

$$\mathcal{N}_{\text{FlexLoRA\_Agg}}^t = \left\| \Delta \mathcal{W}_*^t - \Delta \mathcal{W}_{\text{FlexLoRA}}^t \right\|_F$$
$$= \left\| \Delta \mathcal{W}_*^t - \mathbf{U}^t \mathbf{S}^t (\mathbf{V}^t)^\top \right\|_F$$

### E.3 HETLoRA

HETLoRA zero-pads LoRA modules to a uniform rank for aggregation, and then clients truncate for initialization.

**Initialization Noise ($\mathcal{N}_{\text{HETLoRA\_Init}}^t$):** Server has global LoRA modules $\mathbf{A}^{t-1}$ (rank $R$), $\mathbf{B}^{t-1}$ (rank $R$). Client $i$ initializes by rank $r_i$ truncation: $\mathbf{A}_i^{t-1} = \mathbf{A}_{[:r_i,:]}^{t-1}$, $\mathbf{B}_i^{t-1} = \mathbf{B}_{[:,:r_i]}^{t-1}$. The initialization noise is formulated as,

$$\mathcal{N}_{\text{HETLoRA\_Init}}^t = \sum_{i \in [v]} \left( \left\| \mathbf{A}_{[r_i+1:R,:]}^{t-1} \right\|_F + \left\| \mathbf{B}_{[:,r_i+1:R]}^{t-1} \right\|_F \right)$$

**Aggregation Noise ($\mathcal{N}_{\text{HETLoRA\_Agg}}^t$):** Client $i$ sends $\mathbf{A}_i^t$ (rank $r_i$), $\mathbf{B}_i^t$ (rank $r_i$). Server zero-pads to rank $R$: $\mathbf{A}_i^{t,\prime}, \mathbf{B}_i^{t,\prime}$. Global LoRA modules are averages of padded matrices: $\mathbf{A}^t = \frac{1}{v} \sum_{j \in [v]} \mathbf{A}_j^{t,\prime}$,

$\mathbf{B}^t = \frac{1}{v} \sum_{j \in [v]} \mathbf{B}_j^{t,\prime}$. Actual update: $\Delta \mathcal{W}_{\text{HETLoRA}}^t = \mathbf{B}^t \mathbf{A}^t = \frac{1}{v^2} \sum_{j \in [v]} \sum_{k \in [v]} \mathbf{B}_j^{t,\prime} \mathbf{A}_k^{t,\prime}$. Ideal update: $\Delta \mathcal{W}_*^t = \frac{1}{v} \sum_{l \in [v]} \mathbf{B}_l^t \mathbf{A}_l^t = \frac{1}{v} \sum_{l \in [v]} \mathbf{B}_l^{t,\prime} \mathbf{A}_l^{t,\prime}$. The aggregation noise is:

$$
\begin{aligned}
\mathcal{N}_{\text{HETLoRA\_Agg}}^t &= \left\| \Delta \mathcal{W}_*^t - \Delta \mathcal{W}_{\text{HETLoRA}}^t \right\|_F \\
&= \left\| \frac{1}{v} \sum_{l \in [v]} \mathbf{B}_l^{t,\prime} \mathbf{A}_l^{t,\prime} - \frac{1}{v^2} \sum_{j \in [v]} \sum_{k \in [v]} \mathbf{B}_j^{t,\prime} \mathbf{A}_k^{t,\prime} \right\|_F \\
&= \frac{1}{v^2} \left\| v \sum_{l \in [v]} \mathbf{B}_l^{t,\prime} \mathbf{A}_l^{t,\prime} - \sum_{j \in [v]} \sum_{k \in [v]} \mathbf{B}_j^{t,\prime} \mathbf{A}_k^{t,\prime} \right\|_F \\
&= \frac{1}{v^2} \left\| v \sum_{l \in [v]} \mathbf{B}_l^{t,\prime} \mathbf{A}_l^{t,\prime} - \left( \sum_{l \in [v]} \mathbf{B}_l^{t,\prime} \mathbf{A}_l^{t,\prime} + \sum_{j \in [v], j \neq k} \mathbf{B}_j^{t,\prime} \mathbf{A}_k^{t,\prime} \right) \right\|_F \\
&= \frac{1}{v^2} \left\| (v-1) \sum_{l \in [v]} \mathbf{B}_l^{t,\prime} \mathbf{A}_l^{t,\prime} - \sum_{j \in [v], j \neq k} \mathbf{B}_j^{t,\prime} \mathbf{A}_k^{t,\prime} \right\|_F
\end{aligned}
$$

### E.4 FED-PLORA

Client $i$ updates its selected modules $\{\mathbf{A}_{i,(j)}^t, \mathbf{B}_{i,(j)}^t\}_{j \in \mathcal{K}_i^t}$ and sends them to the server. The server aggregates each $j$-th parallel module independently. Let $\mathcal{Q}_{(j)}^t = \{i \mid i \in [v], j \in \mathcal{K}_i^t\}$ be the set of clients that trained the $j$-th module. The server computes the average for each PLoRA module's $\mathbf{A}$ and $\mathbf{B}$ matrices:

$$
\overline{\mathbf{A}}_{(j)}^t = \frac{1}{|\mathcal{Q}_{(j)}^t|} \sum_{i \in \mathcal{Q}_{(j)}^t} \mathbf{A}_{i,(j)}^t, \quad \overline{\mathbf{B}}_{(j)}^t = \frac{1}{|\mathcal{Q}_{(j)}^t|} \sum_{i \in \mathcal{Q}_{(j)}^t} \mathbf{B}_{i,(j)}^t
$$

The actual global update from Fed-PLoRA is the sum of the products of these averaged components:

$$
\begin{aligned}
\Delta \mathcal{W}_{\text{Fed-PLoRA}}^t &= \sum_{j=1}^R \overline{\mathbf{B}}_{(j)}^t \overline{\mathbf{A}}_{(j)}^t \\
&= \sum_{j=1}^R \left( \frac{1}{|\mathcal{Q}_{(j)}^t|} \sum_{i_1 \in \mathcal{Q}_{(j)}^t} \mathbf{B}_{i_1,(j)}^t \right) \left( \frac{1}{|\mathcal{Q}_{(j)}^t|} \sum_{i_2 \in \mathcal{Q}_{(j)}^t} \mathbf{A}_{i_2,(j)}^t \right) \\
&= \sum_{j=1}^R \frac{1}{|\mathcal{Q}_{(j)}^t|^2} \sum_{i_1 \in \mathcal{Q}_{(j)}^t} \sum_{i_2 \in \mathcal{Q}_{(j)}^t} \mathbf{B}_{i_1,(j)}^t \mathbf{A}_{i_2,(j)}^t
\end{aligned}
$$

The optimal aggregation in this context, as defined for Fed-PLoRA, is an average of client products per parallel module path:

$$
\Delta \mathcal{W}_*^t = \sum_{j=1}^R \left( \frac{1}{|\mathcal{Q}_{(j)}^t|} \sum_{k \in \mathcal{Q}_{(j)}^t} \mathbf{B}_{k,(j)}^t \mathbf{A}_{k,(j)}^t \right)
$$

The aggregation noise is then the difference between these two global updates. We use the formulation consistent with the main paper, which is $\|\Delta \mathcal{W}^t_* - \Delta \mathcal{W}^t_{\text{Fed-PLoRA}}\|_F$:

$$
\mathcal{N}^t_{\text{Fed-PLoRA\_Agg}} = \left\| \Delta \mathcal{W}^t_* - \Delta \mathcal{W}^t_{\text{Fed-PLoRA}} \right\|_F
$$

$$
= \left\| \sum_{j=1}^R \frac{1}{|\mathcal{Q}^t_{(j)}|^2} \sum_{i_1 \in \mathcal{Q}^t_{(j)}} \sum_{i_2 \in \mathcal{Q}^t_{(j)}} \mathbf{B}^t_{i_1,(j)} \mathbf{A}^t_{i_2,(j)} - \sum_{j=1}^R \left( \frac{1}{|\mathcal{Q}^t_{(j)}|} \sum_{k \in \mathcal{Q}^t_{(j)}} \mathbf{B}^t_{k,(j)} \mathbf{A}^t_{k,(j)} \right) \right\|_F
$$

$$
= \left\| \sum_{j=1}^R \frac{1}{|\mathcal{Q}^t_{(j)}|^2} \left( \sum_{i_1 \in \mathcal{Q}^t_{(j)}} \sum_{i_2 \in \mathcal{Q}^t_{(j)}} \mathbf{B}^t_{i_1,(j)} \mathbf{A}^t_{i_2,(j)} - |\mathcal{Q}^t_{(j)}| \sum_{k \in \mathcal{Q}^t_{(j)}} \mathbf{B}^t_{k,(j)} \mathbf{A}^t_{k,(j)} \right) \right\|_F
$$

Let $T'_j$ be the term inside the parenthesis for module $j$:

$$
T'_j = \sum_{i_1 \in \mathcal{Q}^t_{(j)}} \sum_{i_2 \in \mathcal{Q}^t_{(j)}} \mathbf{B}^t_{i_1,(j)} \mathbf{A}^t_{i_2,(j)} - |\mathcal{Q}^t_{(j)}| \sum_{k \in \mathcal{Q}^t_{(j)}} \mathbf{B}^t_{k,(j)} \mathbf{A}^t_{k,(j)}
$$

We can expand the double summation:

$$
\sum_{i_1 \in \mathcal{Q}^t_{(j)}} \sum_{i_2 \in \mathcal{Q}^t_{(j)}} \mathbf{B}^t_{i_1,(j)} \mathbf{A}^t_{i_2,(j)} = \sum_{k \in \mathcal{Q}^t_{(j)}} \mathbf{B}^t_{k,(j)} \mathbf{A}^t_{k,(j)} + \sum_{\substack{i_1,i_2 \in \mathcal{Q}^t_{(j)} \\ i_1 \neq i_2}} \mathbf{B}^t_{i_1,(j)} \mathbf{A}^t_{i_2,(j)}
$$

Substituting this back into $T'_j$:

$$
T'_j = \left( \sum_{k \in \mathcal{Q}^t_{(j)}} \mathbf{B}^t_{k,(j)} \mathbf{A}^t_{k,(j)} + \sum_{\substack{i_1,i_2 \in \mathcal{Q}^t_{(j)} \\ i_1 \neq i_2}} \mathbf{B}^t_{i_1,(j)} \mathbf{A}^t_{i_2,(j)} \right) - |\mathcal{Q}^t_{(j)}| \sum_{k \in \mathcal{Q}^t_{(j)}} \mathbf{B}^t_{k,(j)} \mathbf{A}^t_{k,(j)}
$$

$$
= \sum_{\substack{i_1,i_2 \in \mathcal{Q}^t_{(j)} \\ i_1 \neq i_2}} \mathbf{B}^t_{i_1,(j)} \mathbf{A}^t_{i_2,(j)} - \left( |\mathcal{Q}^t_{(j)}| - 1 \right) \sum_{k \in \mathcal{Q}^t_{(j)}} \mathbf{B}^t_{k,(j)} \mathbf{A}^t_{k,(j)}
$$

So the aggregation noise can also be expressed by substituting this form of $T'_j$:

$$
\mathcal{N}^t_{\text{Fed-PLoRA\_Agg}} = \left\| \sum_{j=1}^R \frac{1}{|\mathcal{Q}^t_{(j)}|^2} \left( \sum_{\substack{i_1,i_2 \in \mathcal{Q}^t_{(j)} \\ i_1 \neq i_2}} \mathbf{B}^t_{i_1,(j)} \mathbf{A}^t_{i_2,(j)} - \left( |\mathcal{Q}^t_{(j)}| - 1 \right) \sum_{k \in \mathcal{Q}^t_{(j)}} \mathbf{B}^t_{k,(j)} \mathbf{A}^t_{k,(j)} \right) \right\|_F
$$

This noise term $T'_j$ (and thus the overall noise) becomes zero if, for each module $j$, the condition $\frac{1}{|\mathcal{Q}^t_{(j)}|} \sum_{k \in \mathcal{Q}^t_{(j)}} (\mathbf{B}^t_{k,(j)} - \overline{\mathbf{B}}^t_{(j)})(\mathbf{A}^t_{k,(j)} - \overline{\mathbf{A}}^t_{(j)}) = \mathbf{0}$ holds, which typically occurs when client updates $\{\mathbf{B}^t_{k,(j)}, \mathbf{A}^t_{k,(j)}\}$ for a given module $j$ are highly similar or aligned across clients $k \in \mathcal{Q}^t_{(j)}$.

# F  ADDITIONAL STUDIES

## F.1  EFFICIENCY OF RANK-BASED FEDERATED FINE-TUNING METHOD

In this section, we demonstrate the efficiency of the rank-based method for resource-constrained devices by analyzing the FLOPs of LoRA modules, overall training throughput, and communication costs under different rank settings. We use the LLaMA-3.1-8B model on the Finance dataset as a representative case (detailed dataset description is shown in Section C.1), with a hidden size of $8192$, a batch size of $4$, and assuming float32 precision, where each parameter occupies 4 bytes.

To compute the FLOPs introduced by LoRA modules, we begin with the forward computation $z = BAx$, which requires $2r(d_{\text{in}} + d_{\text{out}})$ operations per token, where $r$ is the LoRA rank and $d_{\text{in}} = d_{\text{out}} =$

Figure 9: Visualization on the efficiency of the rank-based method.

$d = 8192$ is the hidden dimension. Since the backward pass roughly doubles the compute cost, the total per-token FLOPs for each LoRA module becomes $6rd$. The total LoRA FLOPs per training step can thus be estimated as $6rd \times$ batch size $\times$ sequence length $\times$ number of LoRA modules. For instance, with a sequence length of 512, 32 transformer layers, 2 target modules per layer, result in total 64 LoRA modules, the total becomes $6 \times r \times 8192 \times 4 \times 512 \times 64 = 196,608 \times r$ FLOPs. This value increases linearly with rank. On the training dataset, in order to train one epoch with the full dataset, it requires approximately 10,000 steps with this batch size (i.e., 4), such that the total FLOPs introduced by LoRA modules under rank settings $r = 8, 2$, and 1 are respectively $1.57 \times 10^6 \times 10,000 = 1.57 \times 10^{10}$, $3.93 \times 10^5 \times 10,000 = 3.93 \times 10^9$, and $1.97 \times 10^5 \times 10,000 = 1.97 \times 10^9$ FLOPs per epoch for the LLaMA-3.1-8B model. We visualize the FLOPs reductions for three levels of rank settings in Figure 9 (a).

To assess the training efficiency under different LoRA rank settings, we measure the throughput in terms of seconds per 100 tokens. The actual wall-clock times for the whole training process in our setting are 3.07, 2.84, and 2.41 days under rank settings $r = 8, 2$, and 1, respectively. The experimental and hardware settings are reported in Sections C.1, C.2. This results in throughput values of approximately 0.864, 0.799, and 0.678 seconds per 100 tokens, respectively. These measurements demonstrate that reducing the rank leads to improved computational efficiency during training. We visualize the throughput increase for three levels of rank settings in Figure 9 (b).

To estimate the communication volume for one training epoch, we assume that only the LoRA parameters are transmitted every 10 steps. With 10,000 steps per epoch and transmission frequency every 10 steps, there are 1,000 transmissions in total. Each transmission involves sending and receiving $128rd$ LoRA module parameters, where $r$ is the LoRA rank and $d = 8192$ is the hidden size. Assuming `float32` precision (4 bytes per parameter), this yields a total communication volume of $512,000rd$ bytes per epoch. For ranks $r = 1, 2$, and 8, the resulting communication costs are approximately 3.91 GB, 7.81 GB, and 31.25 GB, respectively. We visualize the communication volume reductions for three levels of rank settings in Figure 9 (c).

From the resource consumption perspective, the rank-based method provides a practical solution for efficient federated fine-tuning of LLMs, as evidenced by the reduction in LoRA module FLOPs and communication volume, as well as improved training throughput. Although the rank-based approach is not specifically designed for memory savings, our method is orthogonal to a wide range of existing memory-efficient techniques and can be easily combined with them (as discussed in Section B).

These resource consumption reductions make low-rank configurations particularly practical for deployment in heterogeneous federated settings. However, as shown in Figure 9 (d), we provide an average performance comparison on Finance datasets under three levels of rank settings. The average accuracies are 52.19%, 51.04%, and 50.05% for ranks $r = 8, 2$, and 1, respectively. This demonstrates that different LoRA rank settings lead to a noticeable performance gap, where even a 1% drop in accuracy is considered significant and typical in LLM instruction fine-tuning tasks on domain-specific downstream datasets. This observation motivates the design of our proposed method, Fed-PLoRA, which aims to incorporate clients experiencing degraded performance under low-rank configurations into a heterogeneous-rank federated training process and improve the overall performance of the global model.

F.2 COMPUTATION, COMMUNICATION, AND MEMORY OVERHEAD OF FED-PLORA

We analyze numerical resource overhead in this section. Here, we use HETLoRA Cho et al. (2024) as a reference to calculate overhead, since it is the naive extension of FedAvg to heterogeneous resource settings. We provide the results as following:

**Computation Overhead.** During the FL process, the computation overhead can be analyzed in the steps of local parameter initialization, local training, and server-side model aggregation Wu & Wang (2022); Zhou et al. (2022). To ground the discussion, we report the per-client, per-round computational workloads under BERT-base Devlin et al. (2019) with PF16 precision. The results (in terms of FLOPs) are reported using 100 clients with a global rank of $R=16$ and a local rank configuration of $\{1, 4, 16\}$, representing three tiers of resource heterogeneity. The number of clients in each tier are 34, 33, and 33, respectively. Each client is assigned 100 training samples. The result are summarized in Table 13. Note that, because local initialization, local training, and local overhead vary across clients, we report the results for the weakest client with $r_i = 1$, as it represents the most resource-constrained setting.

| Method | Local Initialization (FLOPs) | Local Training (FLOPs) | Model Aggregation (FLOPs) | Local Overhead (FLOPs) | Server Overhead (FLOPs) |
|---|---|---|---|---|---|
| HETLoRA | $3.68 \times 10^4$ | $1.45 \times 10^{14}$ | $2.94 \times 10^7$ | - | - |
| FLoRA | $6.26 \times 10^5$ | $1.45 \times 10^{14}$ | $8.88 \times 10^9$ | $+5.89 \times 10^5$ | $+8.85 \times 10^9$ |
| FlexLoRA | $3.68 \times 10^4$ | $1.45 \times 10^{14}$ | $7.71 \times 10^{10}$ | None | $+7.70 \times 10^{10}$ |
| **Fed–PLoRA (Ours)** | $1.41 \times 10^7$ | $1.45 \times 10^{14}$ | $1.10 \times 10^6$ | $+1.40 \times 10^7$ | $-2.83 \times 10^7$ |

Table 13: Computation overhead of Fed-PLoRA compared with SOTA methods.

Overall, the dominant computation cost arises from local training, which on the order of $10^{14}$ FLOPs. Our method adds only negligible overhead during local initialization (on the order of $10^7$ FLOPs), and reduces computational cost for model aggregation compared to HETLoRA.

**Communication Overhead.** The communication cost accounts for both uplink and downlink traffic. Here, we report the per-client, per-round uplink and downlink volumes. The measurements are obtained using FP16 BERT-base Devlin et al. (2019) with a global rank of $R=16$. In Table 14, we report the results for the weakest client with $r_i = 1$, which reflects the largest communication overhead incurred by our method.

| Method | Uplink (MB) | Downlink (MB) | Overhead (MB) |
|---|---|---|---|
| HETLoRA | 0.04 | 0.04 | - |
| FLoRA | 0.04 | 13.54 | +13.50 |
| FlexLoRA | 0.04 | 0.04 | None |
| **Fed–PLoRA (Ours)** | 0.04 | 0.54 | +0.50 |

Table 14: Communication overhead of Fed-PLoRA compared with SOTA methods.

We can see that our method incurs only 0.50 MB of overhead per round for the weakest client, which is negligible even in constrained network settings. We further extend our evaluation to larger models when analyzing the downlink traffic of Fed-PLoRA. As shown in Table 15, in Fed-PLoRA, the downlink traffic is only a few megabytes per round per client, even for models with billions of parameters, making it easily deployable over low-bandwidth networks.

| Model | Downlink (MB) |
|---|---|
| LLaMA-1B | 2.79 |
| OPT-1.3B | 2.62 |
| Mistral-7B-v0.3 | 7.08 |
| Llama-3.1-8B | 7.09 |
| Qwen3-4B-Instruct-2507 | 5.24 |

Table 15: Downlink traffic of Fed–PLoRA across model scales.

| Method | Persistent Base Model (MB) | Persistent Local Trainable Parameters (MB) | Temporary Global Parameters (MB) | Local Training (MB) | Total Overhead (MB) |
|---|---|---|---|---|---|
| HETLoRA | 210.00 | 0.04 | $0.04 \to 0$ | 2717.91 | - |
| FLoRA | 210.00 | 0.04 | $13.54 \to 0$ | 2717.91 | $+13.50 \to 0$ |
| FlexLoRA | 210.00 | 0.04 | $0.04 \to 0$ | 2717.91 | None |
| **Fed–PLoRA (Ours)** | 210.00 | 0.04 | $0.54 \to 0$ | 2717.91 | $+0.50 \to 0$ |

Table 16: Memory overhead of Fed-PLoRA compared with SOTA methods.

**Memory Overhead.** Here, we study the memory usage for the storage of the base model and local trainable LoRA/PLoRA parameters, the temporary memory for the received global parameters during local initialization, and the memory during local training. The results are obtained using FP16 BERT-base Devlin et al. (2019) with a global rank of $R=16$. The batch size for local training is 16, and the input size is 512. We report the memory footprint on the weakest client with rank $r_i = 1$ in Table 16. The persistent base model and local LoRA/PLoRA parameters occupy the same amount of memory across all methods. When dealing with the received global parameters during local initialization, our method requires addtional 0.50 MB of temporary memory, which is negligible compared to the base model's and local training memory costs.

