# OpenReview forum: "Heterogeneous Federated Fine-Tuning with Parallel One-Rank Adaptation"
_ICLR.cc/2026/Conference — ICLR 2026 Poster_

### Official Review · Reviewer_x1y8 · 2025-10-29

**Soundness:** 3
**Presentation:** 4
**Contribution:** 3
**Rating:** 6
**Confidence:** 4

**Summary:**

The paper proposes Fed-PLoRA, a framework for heterogeneous federated fine-tuning (FFT) of large language models. The key idea is to decompose each LoRA module into multiple parallel one-rank modules (PLoRA) and introduce a Select-N-Fold strategy that folds untrained modules into the frozen backbone to mitigate initialization and aggregation noise. The paper provides a unified noise analysis and reports performance gains over prior methods such as FLoRA, FlexLoRA, and HETLoRA on GLUE, Natural Instructions, and other datasets.

**Strengths:**

This paper addresses a timely problem of heterogeneous resource constraints in federated fine-tuning of LLMs. It proposes a modular reformulation, PLoRA, that can in principle generalize to other LoRA-based methods, and introduces a Select-N-Fold strategy to mitigate initialization and aggregation noise. The experiments span multiple models and benchmarks, providing empirical evidence of the proposed method’s efficiency.

**Weaknesses:**

1. The paper overlooks the training FLOPs and storage overhead of the frozen base model on the client side. From my understanding, each client fine-tunes only its LoRA adapters while keeping the base model parameters fixed. However, even though the base model is frozen, it is still fully involved in the forward passes. When the base model is large (e.g., LLaMA-3.1-8B), its forward FLOPs can dominate the total computation, likely exceeding the resource capacity of many local clients. This makes the proposed setup impractical for real-world federated environments.

2. The implementation code is currently unavailable. Releasing it would greatly improve reproducibility and allow the community to validate its efficiency.

**Questions:**

1. Please describe in more detail how the non-IID data distribution is constructed, and the definition of heterogeneity ratio.
2. For a fair comparison, the number of trainable parameters should be explicitly reported in the main experimental results. For instance, what are the trainable parameter counts for each baseline method presented in Table 1, Table 2, and Table 3? This information is critical for interpreting both the performance and communication efficiency.
3. How is the rank $r_i$ for the $i$-th client determined? Can $r_i$ be adaptively assigned according to the complexity of the client’s local task, or it only depends on the client resources.
4. In Table 2, FLoRA performs near random guessing on the CoLA benchmark. Could the authors provide an explanation for this? Even though there exists large initialization noise in the stage of broadcast and initialization.
5. What is the definition of the communication cost in Figure 5, please clarify it or provide the clear reference. Does the communication occur every local training iteration？

---

> ### Author Response · Authors · 2025-11-22
> **Response to Reviewer x1y8 (1/3)**
>
> **We sincerely thank Reviewer x1y8 for the thoughtful and valuable comments. We carefully address each weakness and question in the following responses.**
>
> ---
>
> > [Weakness 1] The paper overlooks the training FLOPs and storage overhead of the frozen base model on the client side. From my understanding, each client fine-tunes only its LoRA adapters while keeping the base model parameters fixed. However, even though the base model is frozen, it is still fully involved in the forward passes. When the base model is large (e.g., LLaMA-3.1-8B), its forward FLOPs can dominate the total computation, likely exceeding the resource capacity of many local clients. This makes the proposed setup impractical for real-world federated environments.
>
> **Response.** Thanks for the questions. Parameter-efficient fine-tuning (PEFT) methods such as LoRA adapt large models by updating only a small subset of parameters, thereby reducing memory usage and backward FLOPs, but not the forward FLOPs of the frozen backbone. This characteristic is shared by all LoRA-based federated fine-tuning methods. As a result, the forward FLOPs of the base model are identical across FedIT, HETLoRA, FLoRA, FlexLoRA, and our method.
>
> All resource-aware federated fine-tuning methods share the same objective: making local fine-tuning feasible on clients with limited resources. A fundamental prerequisite for feasibility is avoiding out-of-memory (OOM) errors so that each client can at least perform the forward and backward passes. Without satisfying this minimum memory requirement, a client would be unable to participate in any federated fine-tuning, regardless of FLOPs or communication cost. When using LoRA (including FedIT, HETLoRA, FLoRA, FlexLoRA, and our method), clients can reduce memory usage for optimizer states by lowering LoRA ranks. Additionally, orthogonal techniques, such as quantizing the base model to reduce model memory or using activation checkpointing to reduce activation memory can further help meet client-side constraints.
>
> We discussed other complementary approaches in Appendix B under Other Resource Efficient Fine Tuning Pathways.
>
>
>
> ---
>
> > [Weakness 2] The implementation code is currently unavailable. Releasing it would greatly improve reproducibility and allow the community to validate its efficiency.
>
> **Response.** Thanks for the suggestion. We are currently cleaning the code to comply with ICLR requirements and will release it before the end of the rebuttal period.
>
> ---
>
> > [Question 1] Please describe in more detail how the non-IID data distribution is constructed, and the definition of heterogeneity ratio.
>
> **Response.** Thanks for giving us the chance to clarify these concepts.
>
> 1. **Non-IID data distribution settings.** We apply both pathological and Dirichlet partitioning strategies to construct non-IID data distributions. The pathological scheme skews the set of tasks available to each client, while the Dirichlet scheme primarily controls the number of samples per client. We use these strategies differently across datasets because Dirichlet partitioning does not significantly affect convergence on instruction-following tasks.
>    1. For the Natural Instructions dataset, we adopt an extreme pathological setting in which each client receives 20 of the 612 tasks
>    2. For Dolly 15K, each client is assigned data from only 1 of the 7 tasks using pathological partitioning.
>    3. For GLUE, we use Dirichlet partitioning with $\alpha = 0.01$, producing a highly imbalanced sample distribution across clients.
>    4. For all other datasets, we use IID partitioning because no task or category labels are available. These details are discussed in Appendix C.
> 2. **Heterogeneity ratio definition.** The heterogeneity ratio measures the proportion of clients in each resource tier relative to the total number of clients. For example, with 100 clients in total, a ratio of 6:3:1 indicates that 60 clients have low capacity, 30 have medium capacity, and 10 have high capacity. Each capacity tier is assigned a corresponding LoRA rank.
>
> We will include these details in the revised manuscript.

---

> ### Author Response · Authors · 2025-11-22
> **Response to Reviewer x1y8 (2/3)**
>
> > [Question 2 & 5] For a fair comparison, the number of trainable parameters should be explicitly reported in the main experimental results. For instance, what are the trainable parameter counts for each baseline method presented in Table 1, Table 2, and Table 3? This information is critical for interpreting both the performance and communication efficiency.
> What is the definition of the communication cost in Figure 5, please clarify it or provide the clear reference. Does the communication occur every local training iteration？
>
> **Response.** Thanks for the question. We would like to clarify as following.
>
> 1. **Same number of local trainable parameters.** For fairness, all methods compared within the same dataset are configured to use an identical number of local trainable parameters, which scales as $O( (d+k)r_i)$ for each client. Under the settings in Table~4, the average number of local trainable parameters across clients for each model is as follows:
>
> | Method               | Downlink    | Overhead   |
> |-----------------------|-----------------------------------|---|
> | HETLoRA  |  0.04 MB (for $\boldsymbol{A_i^g}$ and $\boldsymbol{B_i^g})$   | -   |
> | FLoRA    | 13.54 MB ($\boldsymbol{W^g_i}$) | +13.50 MB   |
> | FlexLoRA |  0.04 MB (for $\boldsymbol{A_i^g}$ and $\boldsymbol{B_i^g})$  | None  |
> | Fed-PLoRA (Ours) |  0.54 MB (for $\boldsymbol{A^g}$ and $\boldsymbol{B^g}$) | +0.50 MB  |
>
> 2. **Uplink Communication.** Since the number of local trainable parameters is identical across methods, the size of the uploaded model updates per client per round is also the same, resulting in identical uplink (client to server) communication cost.
>
> 3. **Downlink Communication.** For downlink (server to client) communication, however, the number of global LoRA/PLoRA parameters differs across methods, leading to variations in downlink traffic. Using the BERT-base model as an example, the downlink traffic shown below remains small (on the order of a few megabytes) for all methods due to the lightweight nature of LoRA parameters. Compared with FedIT, Fed-PLoRA incurs slightly higher downlink traffic because FedIT sends a global LoRA module of rank $r_i$ to client $i$ (here we report the extreme case when $r_i=1$), whereas Fed-PLoRA sends $R$ rank-1 PLoRA module. Nonetheless, the increase remains minimal at only a few megabytes and is negligible in practical federated learning settings.
>
> 4. **Communication Cost in Figure 5.** For Figure 5, we define the communication cost pre round as the total amount of data exchanged between the server and all participanting clients in each training round, including both uplink and downlink transmissions. The values plotted in Figure 5 represent the cumulative communication cost aggregated over training rounds.
>
>
> | Method               | Downlink Traffic                                                                                                                    | Overhead   |
> |-----------------------|-------------------------------------------------------------------------------------------------------------------------------|------------|
> | FedIT (baseline)         | 0.04 MB (for $\boldsymbol{A_i^g}$ and $\boldsymbol{B_i^g})$                                                                       | --         |
> | HETLoRA                     | 0.04 MB (for $\boldsymbol{A_i^g}$ and $\boldsymbol{B_i^g})$                                                                       | None       |
> | FLoRA                      | 4.04 MB (for $\{\boldsymbol{A_i^g}\in\mathbb{R}^{100\times k},\,\boldsymbol{B_i^g}\in\mathbb{R}^{d\times 100}\}$ ) | +4.00 MB   |
> | FlexLoRA                   |  0.04 MB (for $\boldsymbol{A_i^g}$ and $\boldsymbol{B_i^g})$                                                                       | None       |
> | Fed-PLoRA (Ours)       | 0.54 MB (for $\boldsymbol{A^g}$ and $\boldsymbol{B^g}$)                                                         | +0.50 MB   |

---

> ### Author Response · Authors · 2025-11-22
> **Response to Reviewer x1y8 (3/3)**
>
> > [Question 3] How is the rank  for the -th client determined? Can  be adaptively assigned according to the complexity of the client’s local task, or it only depends on the client resources.
>
> **Response.**  Thank you for the question. To determine the LoRA rank for each client, we first define three resource tiers (low, medium, and high). We then empirically evaluate performance of LoRA across a broad set of candidate ranks {1,2,4,8,16,32,64} for each task and identify the largest rank that does not lead to over-parameterization. For the high-resource tier, we assign this largest feasible rank to ensure a balanced trade-off between model capacity and available resources. For the mid-resource tier, we assign intermediate ranks such that the average rank of clients remains below half of the high tier’s rank. For the low-resource tier, the rank is fixed at 1. Once the rank for each tier is determined, clients are randomly assigned to resource tiers according to the specified heterogeneity ratio.
>
> ---
>
> > [Question 4] In Table 2, FLoRA performs near random guessing on the CoLA benchmark. Could the authors provide an explanation for this? Even though there exists large initialization noise in the stage of broadcast and initialization.
>
> **Response.** Thanks for the insightful question.
>
> 1. **Early divergence.** In FLoRA, low resource clients (those with $r_i$ = 1) produce highly noisy updates due to the severe dimensional mismatch between their local rank and the global rank. These noisy updates contaminate the aggregated model and easily cause early divergence, as reflected in Figure 5.
> 2. **Consistently poor performance beyond CoLA.** The reviewer’s impression regarding CoLA stems from an outlier result where the Matthews Correlation Coefficient reaches -4.08\%, indicating a slight negative correlation with the ground truth and performance close to random guessing. However, this is not isolated to CoLA. On MRPC, QQP, QNLI, and RTE (all binary classification tasks), FLoRA likewise collapses to near-random accuracy. This consistent pattern highlights the detrimental effect of noisy, diverging model updates in heterogeneous resource settings and explains FLoRA’s unstable performance across tasks.
>
> We hope the above responses clarify all parts of the reviewer’s concern. We will also refine the manuscript accordingly to make these points more explicit.

---

> ### Author Response · Authors · 2025-11-26
> **Follow-Up Regarding Reviewer x1y8’s Comments**
>
> Dear Reviewer x1y8,
>
> Thank you sincerely for your thoughtful suggestion. We have provided a clean and easy to use implementation along with the dependency configuration in the supplementary material, so that reproduction and further exploration can be done smoothly by the community.
>
> We also hope our current responses have addressed your concerns. If there is anything you feel could be clarified or further strengthened, we would be more than happy to continue working on it. We have sufficient bandwidth to improve the work and incorporate any additional guidance you may provide. Thank you again for your constructive input and for helping us improve this manuscript.
>
> Best regards,
>
> Authors

---

> > ### Comment · Reviewer_x1y8 · 2025-11-26
> >
> > Thanks for the authors’ detailed responses. I think most of my concerns about training and communication costs have been addressed. Based on my overall assessment of this paper, I keep my rating.

---

> > > ### Author Response · Authors · 2025-11-26
> > > **Thank you Reviewer x1y8**
> > >
> > > Dear Reviewer x1y8,
> > >
> > > Thank you sincerely for your positive feedback. We are pleased to know that our responses addressed your concerns. We truly appreciate your time and constructive suggestions, which have contributed meaningfully to improving the work. We will incorporate these revisions into the final manuscript.
> > >
> > > Best regards,
> > >
> > > Authors

---

### Official Review · Reviewer_AmKz · 2025-10-31

**Soundness:** 4
**Presentation:** 4
**Contribution:** 3
**Rating:** 6
**Confidence:** 5

**Summary:**

This paper tackles the critical challenge of heterogeneous LoRA ranks in Federated Fine-Tuning (FFT). The authors identify that current methods suffer from initialization noise and aggregation noise due to rank mismatches. Their proposed solution, Fed-PLORA, uses PLORA (Parallel One-Rank Adaptation) to re-parameterize LoRA modules as a sum of parallel rank-1 components . This enables a novel Select-N-Fold strategy, where resource-constrained clients train a random subset of modules and "fold" the rest into the frozen weights. This method is theoretically shown to achieve zero initialization noise and is empirically demonstrated to consistently outperform existing heterogeneous FFT methods like FLORA, HETLORA, and FlexLoRA.

**Strengths:**

- **Clear and Compelling Motivation:** The paper is exceptionally well-motivated. It formalizes the *specific failure modes* of prior art: initialization noise and aggregation noise. The entire paper is a clear and focused effort to solve these two problems.
- **Strong Theoretical Analysis:** The noise analysis theoretically proves that the proposed Fed-PLORA framework eliminates initialization noise and provides a powerful and fundamental justification for the method's design.
- **Simple and Effective Methodology:** The proposed solution is both elegant and practical.
- **Comprehensive Empirical Validation:** The experimental results are strong and thorough. The authors test on a wide variety of models and diverse datasets.

**Weaknesses:**

- Adding pseudocode for the algorithm would improve clarity.
- It appears that PLoRA requires downloading the entire global LoRA model. In contrast, other methods—if rank is publicly available—can use much smaller downloads. Although Section 4.2 addresses this, the R − ri downlink cost could be quite high when R is large and ri is small, potentially causing synchronization issues.

**Questions:**

PLoRA folds the remaining R − ri untrained rank modules into the pretrained weight. Is this different from sparse LoRA tuning where the corresponding R − ri untrained rows are frozen?

---

> ### Author Response · Authors · 2025-11-22
> **Response to Reviewer AmKz (1/2)**
>
> **We are grateful to Reviewer AmKz for the thoughtful and valuable comments. We carefully address each point and question in the following responses.**
>
> >[Weakness 1] Adding pseudocode for the algorithm would improve clarity.
>
>
> **Response.** We appreciate this suggestion. We provide the pseudocode below and will have it in the revised manuscript (We apologize for the formatting, as OpenReview does not fully support standard LaTeX rendering. The reviewer may paste the code into a Markdown editor for clearer viewing.).
>
> ```
> **Algorithm Fed-PLoRA**
>
> **Require:** number of communication rounds $T$, global rank $R$, local ranks $\{r_i\}_{i\in[v]}$, pre-trained backbone $\boldsymbol{\Theta}^0$, initial global PLoRA parameters $\boldsymbol{\theta}^0$
>
> 1. Server sends $\boldsymbol{\Theta}^0$ to all clients
> 2. **For** $t = 0$ to $T - 1$ **do**
>    1. Server samples a set of clients $\mathcal{S}^t \subseteq [v]$
>    2. Server sends global PLoRA parameters $\boldsymbol{\theta}^{t}$ to all $i \in \mathcal{S}^t$
>    3. **For** each client $i \in \mathcal{S}^t$ **in parallel do**
>       1. Randomly sample a subset $\mathcal{K}_i^{t}$ of size $r_i$ from index set $[R]$
>       2. Initialize local trainable PLoRA modules: $\theta_i^t \leftarrow \{(A_j^t, B_j^t)\;|\; j \in \mathcal{K}_i^t\}$
>       3. Fold unselected modules into the frozen target weights: $\mathcal{W}_i^{t} \leftarrow \mathcal{W}^{0} + \sum_{j \notin \mathcal{K}_i^{t}} B_{(j)}^{t} A_{(j)}^{t}$
>          to obtain the local frozen backbone $\Theta_i^{t}$
>       4. Compute local update: $\boldsymbol{\delta}_i^{t} \leftarrow$ LocalUpdate$(\boldsymbol{\theta}_i^{t}, D_i, \boldsymbol{\Theta}_i^t)$
>    4. **End for**
> 3. Rank wise aggregation of PLoRA modules: $\boldsymbol{\theta}^{t+1} \leftarrow$ Agg$\big(\{\boldsymbol{\delta}_i^{t}\}_{i\in\mathcal{S}^t}, \{\mathcal{K}_i^{t}\}_{i\in\mathcal{S}^t}\big)$
> 4. Update global model: $\boldsymbol{\Phi}_g^{t+1} \leftarrow \{\boldsymbol{\Theta}^0, \boldsymbol{\theta}^{t+1}\}$
> 5. **End for**
>
> **Return** $\boldsymbol{\Phi}_g^T=\{\boldsymbol{\Theta}^0, \boldsymbol{\theta}^T \}$
> ````
> ---
>
> >[Weakness 2] It appears that PLoRA requires downloading the entire global LoRA model. In contrast, other methods—if rank is publicly available—can use much smaller downloads. Although Section 4.2 addresses this, the R − ri downlink cost could be quite high when R is large and ri is small, potentially causing synchronization issues.
>
> **Response.** Thanks for the question. We address these concerns from two perspectives.
> 1. Since all the methods (including our framework) are built on parameter-efficient fine-tuning, the global LoRA module remains small in practice.
>
> 2. In Fed-PLoRA, each rank-1 module is highly lightweight (two vectors of dimensions $d$ and $k$), and the total downlink payload scales as ${O}((d+k)R)$. Below, we report the per-client downlink traffic of Fed-PLoRA across different models with $R=16$. As shown, the traffic is only a few megabytes, which is acceptable in practical federated settings. The additional downlink overhead relative to FedIT scales as ${O}((d+k)(R-r_i)$ for each client. We report the worst-case overhead, which occurs for the least capable client with $r_i=1$. Because the downlink traffic itself is small, the resulting overhead is also negligible.
>
>     **Downlink traffic and overhead when using different models ($R=16$, 16-bit precision):**
>
>     | Model| Downlink Traffic |  Downlink Overhead (worst case)  |
>     |----|---|----|
>     | BERT-base| 0.54 MB|  0.50 MB |
>     | LLaMA-1B | 2.79 MB  | 2.62 MB |
>     | OPT-1.3B | 2.62 MB | 2.46 MB |
>     | Mistral-7B-v0.3 | 7.08 MB | 6.64 MB |
>     | Llama-3.1-8B| 7.09 MB| 6.65 MB  |
>     | Qwen3-4B-Instruct-2507  | 5.24 MB  | 4.91 MB   |
>     | Qwen3-30B-A3B | 5.58 MB | 5.24 MB |
>     | Qwen3-235B-A22B-Instruct-2507   | 21.97 MB | 20.60 MB  |

---

> ### Author Response · Authors · 2025-11-22
> **Response to Reviewer AmKz (2/2)**
>
> >[Question 1] PLoRA folds the remaining R − ri untrained rank modules into the pretrained weight. Is this different from sparse LoRA tuning where the corresponding R − ri untrained rows are frozen?
>
> **Response.** Thanks for the insightful question. We would like to answer this question in the following three aspects.
>
> 1. **Same training dynamics.** Theoretically, freezing the untrained rows should yield the same effect during the local training phase.
>
> 2. **GPU overhead.** Simply freezing the untrained rows to avoid updating increases memory consumption.
>    1. The additional $R − r_i$ rows must still be stored in GPU memory.
>    2. Moreover, these frozen rows still participate in the forward pass, which increases computational cost and peak memory usage because the corresponding activations must also be computed and stored. Folding the untrained rows into the base model avoids these extra computational and memory costs.
>
> 3. **Limited hardware support.** True sparse LoRA training is not widely supported in current software–hardware stacks. Existing hardware acceleration typically requires strictly structured sparsity patterns. For example, NVIDIA A100 Sparse Tensor Cores accelerate only the 2:4 structured sparsity pattern, where exactly two out of every four contiguous weights must be zero [1]. Simply freezing arbitrary rows (e.g., “freezing $R - r_i$ out of $R$ rows”) does not impose such structure and therefore cannot activate hardware sparsity acceleration or provide any GPU speedup.
>
> [1] LEARNING N:M FINE-GRAINED STRUCTURED SPARSE NEURAL NETWORKS FROM SCRATCH, ICLR 2021.
>
> We hope this explanation fully addresses the reviewer’s questions. We will ensure that these details are clearly stated in the updated manuscript.

---

> ### Comment · Reviewer_AmKz · 2025-11-26
>
> I appreciate the authors’ response. My concerns are fully addressed, and I will raise my score.

---

> > ### Author Response · Authors · 2025-11-26
> > **Thank you Reviewer AmKz**
> >
> > Dear Reviewer AmKz,
> >
> > We sincerely thank you for your positive feedback and for raising the score. We are glad that our responses have addressed your concerns. We appreciate your time and constructive comments, which have helped improve the quality of our work. We will ensure that this information is incorporated into the final manuscript.
> >
> > Best regards,
> >
> > Authors

---

### Official Review · Reviewer_G2mv · 2025-11-01

**Soundness:** 3
**Presentation:** 3
**Contribution:** 3
**Rating:** 6
**Confidence:** 4

**Summary:**

This paper targets a problem of resource heterogeneity in Federated Learning (FL) for fine-tuning LLMs. A common solution for federated fine tuning for clients with different computational resources is to adapt LoRA modules of different ranks. The authors argue that this heterogeneity introduces two primary issues: initialization noise (when low-resource clients must truncate or discard parts of the global model) and aggregation noise (when the server attempts to combine modules of different dimensions).

To solve this, the paper proposes Fed-PLORA, a novel framework built on two core ideas: 1) Parallel One-Rank Adaptation  or PLORA where instead of having a standard rank-R LoRA module, they propose utilizing the  sum of R parallel one-rank modules. 2) Select-N-Fold Strategy which is A new initialization and training protocol.

The authors claim that the "Select-N-Fold" strategy completely eliminates initialization noise, as no information from the global model is
discarded. They analyze the remaining aggregation noise and argue it is minimal. Through extensive experiments, they demonstrate that Fed-PLORA outperforms existing heterogeneous FFT methods.

**Strengths:**

* The paper is well-written and the authors did a good job explaining the existing problems.

* Through various settings and different empirical results the authors show the merits of their algorithms.

* The authors did a proper ablation study, explaining the importance of each component.

**Weaknesses:**

One important aspect of the paper is the Downlink Communication cost. The "Select-N-Fold" strategy has one clear limitation that is understated: downlink communication cost.

**Questions:**

Can you make a table (at least for one setting) to show all the new costs of your method and compare with the prior works?

---

> ### Author Response · Authors · 2025-11-22
> **Response to Reviewer G2mv (1/2)**
>
> **We sincerely appreciate Reviewer G2mv’s constructive feedback. We respond to all concerns in detail below.**
>
> ---
>
> > [Weaknesses&Questions] One important aspect of the paper is the Downlink Communication cost. The "Select-N-Fold" strategy has one clear limitation that is understated: downlink communication cost. Can you make a table (at least for one setting) to show all the new costs of your method and compare with the prior works?
>
> **Response.** Thank you for highlighting this important point.
> 1. **Overall Overhead Comparisons.** To fully address the reviewer’s concern, we report the communication, computation, and memory overhead per client per round separately below, in comparison with FedIT. We pick the setting using BERT-base model with FP16 precision, 100 clients, local rank $r_i = 1$ (corresponding to the least capable client), and global rank $R = 16$.
>
> **Communication Overhead:**
>
> | Method| Uplink  |Downlink   | Overhead   |
> |------|-----|-----|----|
> | HETLoRA  | 0.04 MB (for $\boldsymbol{A_i}$ and $\boldsymbol{B_i}$)  | 0.04 MB (for $\boldsymbol{A_i^g}$ and $\boldsymbol{B_i^g})$   | -   |
> | FLoRA    | 0.04 MB (for $\boldsymbol{A_i}$ and $\boldsymbol{B_i}$)  | 13.54 MB ($\boldsymbol{W^g_i}$) | +13.50 MB   |
> | FlexLoRA | 0.04 MB (for $\boldsymbol{A_i}$ and $\boldsymbol{B_i}$)  |  0.04 MB (for $\boldsymbol{A_i^g}$ and $\boldsymbol{B_i^g})$  | None  |
> | Fed-PLoRA (Ours) | 0.04 MB (for $\boldsymbol{A_i}$ and $\boldsymbol{B_i}$) | 0.54 MB (for $\boldsymbol{A^g}$ and $\boldsymbol{B^g}$) | +0.50 MB  |
>
> **Computational Overhead:**
>
> | Method  |  Local Initialization   |Local Training | Model Aggregation   | Local Overhead  | Server Overhead|
> |-----|---------|------|------|-----|----|
> | HETLoRA  |  $3.68\times10^{4}$ FLOPs (for updating local LoRA)                                     | $1.45\times10^{14}$ FLOPs (for forward and backward) |$2.94\times10^{7}$ FLOPs (for padding and truncation)        | - | -|
> | FLoRA  |  $6.26\times10^{5}$ FLOPs (for updating local target module)  | $1.45\times10^{14}$ FLOPs (for forward and backward) |$8.88\times10^{9}$ FLOPs (for concatenation local updates and constructing and updating full target module $\boldsymbol{W}_i$)                 | $+5.89\times10^{5}$ FLOPs | $+8.85\times10^{9}$ FLOPs
> | FlexLoRA|  $3.68\times10^{4}$ FLOPs (for updating local LoRA)   | $1.45\times10^{14}$ FLOPs (for forward and backward) |$7.71\times10^{10}$ FLOPs (for averaging and SVD)         | None | $+7.70\times10^{10}$ FLOPs
> | Fed-PLoRA (Ours)  |  $1.41\times10^{7}$ FLOPs (for select-N-fold and updating local LoRA)  | $1.45\times10^{14}$ FLOPs (for forward and backward) | $1.10\times10^{6}$ FLOPs (for rank-wise averaging)   | $+1.41\times10^{7}$ FLOPs | $-2.83\times10^{7}$

---

> ### Author Response · Authors · 2025-11-22
> **Response to Reviewer G2mv (2/2)**
>
> **Memory Overhead during Local Initialization:**
>
> | Method| Persistent Base Model|Persistent Local Trainable Parameters| Temporary Global Parameters|Local Training| Total Overhead|
> |---|---|---|---|---|---|
> | HETLoRA| 210.00 MB (for $\boldsymbol{\Theta}^0$)          | 0.04 MB (for $\boldsymbol{A_i}$ and $\boldsymbol{B_i}$)  | 0.04 MB → 0 MB ( for $\boldsymbol{A_i^g}$ and $\boldsymbol{B_i^g}$) | 2717.91 MB (for activation, optimization, others) | -                        |
> | FLoRA|210.00 MB (for $\boldsymbol{\Theta}^0$)        | 0.04 MB (for $\boldsymbol{A_i}$ and $\boldsymbol{B_i}$) | 13.54 MB → 0 MB (for $\boldsymbol{W}^g_i$)  | 2717.91 MB (for activation, optimization, others)| +13.50 MB → 0 MB   |
> | FlexLoRA|210.00 MB (for $\boldsymbol{\Theta}^0$)       | 0.04 MB (for $\boldsymbol{A_i}$ and $\boldsymbol{B_i}$)  | 0.04 MB → 0 MB (for $\boldsymbol{A_i^g}$ and $\boldsymbol{B_i^g}$ )                                                            | 2717.91 MB (for activation, optimization, others)   | None                         |
> | Fed-PLoRA (ours)|210.00 MB (for $\boldsymbol{\Theta}^0$)   |0.04 MB (for $\boldsymbol{A_i}$ and $\boldsymbol{B_i}$) | 0.54 MB → 0 MB (for $\boldsymbol{A}^g$ and $\boldsymbol{B}^g$ )                               | 2717.91 MB (for activation, optimization, others)        | +0.50 MB → 0 MB   |
>
> To measure the temporary memory of FLoRA, we assume 100 participating clients. In all of these federated fine-tuning setups, clients are assumed to have sufficient resources to fine tune at least rank-1 LoRA modules locally. Note that all methods incur identical memory usage during local training, due to having the same number of trainable parameters and an (theoretically) identical LoRA structure.
>
> Specifically, this comparison makes clear that Select-N-Fold introduces only a small additional downlink traffic (0.50 MB per round for BERT-base) and a small temporary memory buffer (0.54 MB), while maintaining comparable uplink cost and significantly lower computation cost than FLoRA and FlexLoRA.
>
> 2. **Additional Downlink Overhead Results.** We additionally report the downlink overhead of our method relative to FedIT across evaluated models of varying scales. Across BERT-base, LLaMA-1B, OPT-1.3B, Mistral-7B, and Llama-3.1-8B models, the downlink overhead ranges from 0.50 MB to 6.65 MB, which remains manageable in realistic federated settings.
>
>     | Model | Downlink Overhead (16-bit precision) |
>     |-----|----|
>     | BERT-base| 0.50 MB|
>     | LLaMA-1B | 2.62 MB  |
>     | OPT-1.3B | 2.46 MB  |
>     | Mistral-7B-v0.3 | 6.64 MB|
>     | Llama-3.1-8B | 6.65 MB |
>
> We hope this answers the reviewer’s concern, and we will add these clarifications clearly in the revision.

---

### Official Review · Reviewer_HFN8 · 2025-11-04

**Soundness:** 3
**Presentation:** 3
**Contribution:** 2
**Rating:** 4
**Confidence:** 1

**Summary:**

Fed-PLoRA addresses federated fine-tuning with heterogeneous LoRA ranks by proposing a new framework to mitigate both issues where rank mismatches introduce substantial initialization and aggregation noise. It introduces PLoRA, which replaces a single rank-R LoRA module with R parallel rank-1 modules that are mathematically equivalent to standard LoRA. Combined with a Select-N-Fold strategy, the method achieves zero initialization noise and reduces aggregation noise under heterogeneous budgets. Experiments across multiple LLM fine-tuning tasks (e.g., GLUE and instruction-following) show consistent accuracy gains over FLoRA, FlexLoRA, and HETLoRA, while avoiding the heavy SVD overhead that hurts communication and training time. It’s easy to adopt in practice though broadcasting all R modules does add some downlink cost.

**Strengths:**

1. This paper precisely defines initialization and aggregation noise in heterogeneous LoRA settings and shows Fed-PLoRA removes the former and reduces the latter.
2.  PLoRA’s parallel rank-1 decomposition is mathematically equivalent to standard LoRA yet naturally supports heterogeneity; paired with Select-N-Fold, it guarantees zero initialization noise while curbing aggregation noise.
3. Strong and robust empirical results. Fed-PLoRA consistently outperforms FLoRA/FlexLoRA/HETLoRA across tasks (e.g., GLUE), and remains robust as client counts and rank distributions vary, including challenging non-IID scenarios.

**Weaknesses:**

1. The method broadcasts all R parallel rank-1 modules to every client and asks clients to keep folded modules, downlink traffic and on-device storage could become non-trivial in weak-network or mobile scenarios.
2. The paper’s empirical validation relies on relatively small or outdated and non-unified base models, which limits generalizability; it would be stronger to standardize on modern backbones like Qwen3 and Llama 3.2 across multiple sizes.
3. Its benchmark suite skews toward easier tasks (e.g., GLUE and basic instruction following) and should incorporate more rigorous reasoning/knowledge evaluations such as MMLU-Pro, GPQA, MuSR, MATH, IFEval, and BBH.
4. The authors do not clearly report the untuned base-model performance, obscuring absolute gains.

**Questions:**

See weakness.

---

> ### Author Response · Authors · 2025-11-22
> **Response to Reviewer HFN8 (1/3)**
>
> **We appreciate Reviewer HFN8 for the thoughtful and constructive comments. We provide a detailed response to every concern and question raised below.**
>
> ---
>
> > [Weakness 1] The method broadcasts all R parallel rank-1 modules to every client and asks clients to keep folded modules, downlink traffic and on-device storage could become non-trivial in weak-network or mobile scenarios.
>
> **Response.** We sincerely thank the reviewer for raising this concern. We clarify that broadcasting the $R$ parallel rank-1 modules does not introduce a downlink or storage bottleneck.
>
> 1. **Downlink traffic remains small.**
> In Fed-PLoRA, each rank-1 module is highly lightweight (two vectors of dimensions $d$ and $k$), and the total downlink payload scales as ${O}((d+k)R)$, which remains orders of magnitude smaller than transmitting the full model. Even in mobile or low-bandwidth settings, the resulting downlink traffic (typically a few megabytes per round as shown in the table below) is negiligible.
>
>     **Downlink traffic when using different models ($R=16$, 16-bit precision)**
>
>     | Model | Downlink Traffic |
>     |---|---|
>     | BERT-base | 0.54 MB   |
>     | LLaMA-1B         | 2.79 MB                     |
>     | OPT-1.3B                                       | 2.62 MB                     |
>     | Mistral-7B-v0.3                               | 7.08 MB                     |
>     | Llama-3.1-8B                                  | 7.09 MB                     |
>     | Qwen3-4B-Instruct-2507                | 5.24 MB                     |
>     | Qwen3-30B-A3B                             | 5.58 MB                     |
>     | Qwen3-235B-A22B-Instruct-2507   | 21.97 MB                    |
>
>
>     These values (in MBs) are small in practice. Even under conservative 3G or congested 4G conditions, transfers of this size complete within a few seconds. Large-scale mobile FL deployments (e.g., Gboard) routinely transmit tens of megabytes per round. Therefore, the downlink cost introduced by Fed-PLoRA remains well within realistic bandwidth budgets.
>
> 1. **Persistent storage remains unchanged, and temporary memory usage for the $R$ modules remains small.**
>    1. Across all methods, each client always keeps the base model and its PLoRA/LoRA modules locally for training. In Fed-PLoRA, once the client receives the $R$ parallel rank-1 PLoRA modules, the $(R - r_i)$ frozen modules are immediately folded into the corresponding pre-trained weights in the base model (which is already resident on the GPU). After folding, these modules are deleted to free memory, so no additional persistent storage is required for these PLoRA modules.
>
>    2. All methods incur some temporary memory usage during local initialization. The table below summerizes the memory usage per client for the base model, local LoRA (or PLoRA in our case) and global LoRA/PLoRA, as well as the resulting overhead compared with FedIT.
>
>     **Memory usage of a client when using BERT-base model ($R=16$,  $r_i=1$)**
>
>    | Method| Persistent Base Model|Persistent Local Trainable Parameters| Temporary Global Parameters|Local Training| Total Overhead|
>    |---|---|---|---|---|---|
>    | HETLoRA| 210.00 MB (for $\boldsymbol{\Theta}^0$)          | 0.04 MB (for $\boldsymbol{A_i}$ and $\boldsymbol{B_i}$)  | 0.04 MB → 0 MB ( for $\boldsymbol{A_i^g}$ and $\boldsymbol{B_i^g}$) | 2717.91 MB (for activation, optimization, others) | -                        |
>    | FLoRA|210.00 MB (for $\boldsymbol{\Theta}^0$)        | 0.04 MB (for $\boldsymbol{A_i}$ and $\boldsymbol{B_i}$) | 13.54 MB → 0 MB (for $\boldsymbol{W}^g_i$)  | 2717.91 MB (for activation, optimization, others)| +13.50 MB → 0 MB   |
>    | FlexLoRA|210.00 MB (for $\boldsymbol{\Theta}^0$)       | 0.04 MB (for $\boldsymbol{A_i}$ and $\boldsymbol{B_i}$)  | 0.04 MB → 0 MB (for $\boldsymbol{A_i^g}$ and $\boldsymbol{B_i^g}$ )                                                            | 2717.91 MB (for activation, optimization, others)   | None                         |
>    | Fed-PLoRA (ours)|210.00 MB (for $\boldsymbol{\Theta}^0$)   |0.04 MB (for $\boldsymbol{A_i}$ and $\boldsymbol{B_i}$) | 0.54 MB → 0 MB (for $\boldsymbol{A}^g$ and $\boldsymbol{B}^g$ )                               | 2717.91 MB (for activation, optimization, others)        | +0.50 MB → 0 MB   |
>
>
>     To measure the temporary memory of FLoRA, we assume 100 participating clients. In all of these federated fine-tuning setups, clients are assumed to have sufficient resources to fine tune at least rank-1 LoRA modules locally. The memory overhead introduced by the $R$ parallel rank-1 PLoRA modules remains negligible.

---

> ### Author Response · Authors · 2025-11-22
> **Response to Reviewer HFN8 (2/3)**
>
> > [Weakness 2&3] The paper’s empirical validation relies on relatively small or outdated and non-unified base models, which limits generalizability; it would be stronger to standardize on modern backbones like Qwen3 and Llama 3.2 across multiple sizes.
> Its benchmark suite skews toward easier tasks (e.g., GLUE and basic instruction following) and should incorporate more rigorous reasoning/knowledge evaluations such as MMLU-Pro, GPQA, MuSR, MATH, IFEval, and BBH.
>
> **Response.** Thank you for the insightful suggestion. After careful consideration, we explain our design choices and provide additional results below.
>
>    1. **LLM selection and hardware constraints.** Within the range of models that can be trained on our available GPU server, we selected Bert-base, Llama-1B, OPT-1.3B, Mistral-7B, and Llama-3.1-8B. These models cover diverse architectures, parameter scales, and instruction-following capabilities. Larger modern models (e.g., Qwen3-30B and Qwen3-235B) exceed the memory capacity of our hardware, making fine-tuning infeasible in our capability.
>    2. **Benchmarks beyond GLUE and standard instruction following.** We fully agree that challenging reasoning and knowledge-intensive evaluations are important. In addition to GLUE and standard instruction-following datasets, our evaluation also included domain-specific knowledge tasks in finance (FPB, FIQA, TFNS) and medicine (PubMedQA, MedMCQA, MedQA, CareQA). Following the reviewer’s recommendation, we additionally evaluate on the MATH benchmark [1], which tests algebra, geometry, number theory, combinatorics, and multi-step symbolic reasoning. Detailed results across all subcategories are reported using the Qwen3-4B-A3B-Instruct-2507 model, with $R=8$. We also tested Qwen3-30B, but it ran out of GPU memory.
>
> [1] Measuring Mathematical Problem Solving With the MATH Dataset, NeurIPS 2021.
>
> | Model| Category  | Method | algebra | counting_and_probability | geometry | intermediate_algebra | number_theory | prealgebra | precalculus | Average |
> |---|---|------|-----------|------|------|----------|-------|-------|-----------------|----------|
> | Qwen3-4B-A3B-Instruct-2507     | -     | Untuned Model  | 12.38   | 5.90  | 3.75         | 1.32 | 5.74  | 15.61       | 4.57         | 7.94    |
> | Qwen3-4B-A3B-Instruct-2507     | Homo.   | FedIT (Rank=R)                | 42.20  | 26.79                                | 19.20        | 13.73     | 26.11   | 42.93       | 10.98        | 28.38   |
> | Qwen3-4B-A3B-Instruct-2507     | Homo.   | FedIT (Rank=R/2)             | 36.47  | 22.57                                | 16.70       | 11.62    | 21.48   | 37.08       | 12.27        | 24.62   |
> | Qwen3-4B-A3B-Instruct-2507     | Hete. R=8 (Avg Rank=4) | FLoRA   | 12.72   | 6.75   | 4.38  | 1.88    | 5.37   | 15.04       | 2.93         | 7.94    |
> | Qwen3-4B-A3B-Instruct-2507     | Hete. R=8 (Avg Rank=4) | FlexLoRA   | 38.83   | 19.62   | 15.65        | 9.52    | 22.03  | 40.41       | 10.62        | 24.88   |
> | Qwen3-4B-A3B-Instruct-2507     | Hete. R=8 (Avg Rank=4) | HETLoRA  | 28.47   | 12.23   | 11.69        | 9.85    | 17.77  | 24.79       | 10.07        | 18.16   |
> | Qwen3-4B-A3B-Instruct-2507     | Hete. R=8 (Avg Rank=4) | Fed-PLoRA     | 43.80  | 24.89  | 18.99        | 14.17  | 27.77    | 42.59       | 12.82        | 28.96   |
> | Qwen3-30B-A3B-Instruct-2507    | -    | -   | OOM   | OOM  | OOM         | OOM   | OOM    | OOM         | OOM          | OOM     |
> | Qwen3-235B-A22B-Instruct-2507| -    | -    | OOM   | OOM  | OOM    | OOM   | OOM   | OOM   | OOM  | OOM     |
>
> Across all MATH subcategories on Qwen3-4B, Fed-PLoRA consistently provides reliable gains in algebra, counting, geometry, and other reasoning categories, demonstrating strong robustness under heterogeneous client capabilities. In contrast, methods such as FLoRA and HETLoRA exhibit either collapse or substantial inconsistency when local ranks differ, underscoring their sensitivity to rank heterogeneity. The overall scores appear low because MATH is an exceptionally challenging benchmark in which even small mistakes in multi step exact derivations lead to incorrect final answers.
>
>    3.  Additional reasoning benchmarks suggested by the reviewer (MMLU Pro, GPQA, MuSR, IFEval, and BBH) are evaluation-only datasets and cannot be directly used for federated fine tuning, as clients must perform full supervised updates. For fair comparison, we therefore focus on tasks that support both training and evaluation within the federated fine tuning settings. We remain open to conducting additional evaluations in the federated fine-tuning setting within our computational constraints and would greatly appreciate any further recommendations from the reviewer.

---

> ### Author Response · Authors · 2025-11-22
> **Response to Reviewer HFN8 (3/3)**
>
> > [Weakness 4] The authors do not clearly report the untuned base-model performance, obscuring absolute gains.
>
> **Response.** Thanks for the suggestion. Below, we report the performance of the untuned base model on all evaluated tasks, along with the performance gains achieved by our method over the untuned baseline.
>
> **Results on GLUE:**
>
> | Method        | CoLA | SST-2 | MRPC | QQP  | QNLI | RTE  | Avg   |
> |---------------|------|--------|-------|-------|--------|--------|--------|
> | Untuned Model | 1.20 | 49.08 | 31.61 | 63.09 | 49.95 | 52.70 | 41.27 |
> | Fed-PLoRA     | 59.38 | 92.35 | 86.35 | 87.25 | 88.54 | 65.46 | 79.89 |
> | Gain          | +58.18 | +43.27 | +54.74 | +24.16 | +38.59 | +12.76 | +38.62 |
>
>
> **Results on Natural Instruction:**
>
> | Method        | Natural Instruction (IID) | Natural Instruction (Non-IID) |  Avg   |
> |---------------|----------------------------|-------------------------------|-------|
> | Untuned Model | 33.82                      | 33.82                         |  33.82|
> | Fed-PLoRA     | 64.96                      | 60.76                         |  62.86|
> | Gain          | +31.14                     | +26.94                        | +29.04|
>
> **Results on Financial Datasets:**
> | Method        | FPB   | FIQA  | TFNS  | Avg   |
> |---------------|--------|--------|--------|--------|
> | Untuned Model | 50.57 | 29.33 | 49.87 | 40.12 |
> | Fed-PLoRA     | 63.94 | 31.68 | 64.19 | 53.27 |
> | Gain          | +13.37 | +2.35 | +14.32 | +13.15 |
>
> **Results on Dolly-15K:**
> | Method        | Dolly-15K (IID) | Dolly-15K (Non-IID) |      Avg |
> |---------------|------------------|-----------------------|-------|
> | Untuned Model | 40.02            | 40.02                | 40.02  |
> | Fed-PLoRA     | 60.07            | 59.69                | 59.88  |
> | Gain          | +20.05           | +19.67               | +19.86 |
>
> **Results on Medical Datasets:**
> | Method        | PubMedQA | MedMcQA | MedQA | CareQA | Avg   |
> |---------------|-----------|----------|--------|---------|--------|
> | Untuned Model | 55.77    | 38.43   | 40.92 | 41.28  | 44.10 |
> | Fed-PLoRA     | 70.20    | 41.50   | 46.11 | 46.85  | 51.16 |
> | Gain          | +14.43   | +3.07   | +5.19 | +5.57  | +7.06 |
>
> We hope this clarifies the concerns from the reviewer. We will add these results in the revised manuscript.

---

### Author Response · Authors · 2025-11-26
**Follow-Up for Reviewers**

Dear Reviewers,

Thank you once again for dedicating your time to carefully reviewing our paper and for offering thoughtful suggestions that helped us strengthen the work in many meaningful ways. We truly appreciate the insight and care reflected in each comment. For reviewers whose concerns have been fully addressed, we are grateful for your recognition and support. For any remaining questions or points that may still benefit from clarification, we would be glad to continue the discussion.

Thank you for your time and expertise.

Best regards,

Authors

---

### Author Response · Authors · 2025-12-04
**General Response to ACs and Senior ACs**

Dear Area Chairs and Senior Area Chairs,

We sincerely appreciate your time and effort in meta-reviewing our submission under these special circumstances. We also thank the reviewers for their thoughtful insights and careful evaluations.

Overall, the reviewers recognize our contributions on analyzing initialization and aggregation noise in heterogeneous federated fine-tuning methods (HFN8, G2mv, AmKz, x1y8), the novel design of the PLoRA module (HFN8, AmKz, x1y8), the comprehensive empirical evaluations (HFN8, G2mv, AmKz, x1y8), and the overall quality of the writing (G2mv, AmKz).
During the discussion phase, Reviewers AmKz and x1y8 expressed satisfaction with our responses. **Reviewer AmKz raised the score from 6 to 8, and Reviewer x1y8 maintained a positive evaluation.**
Below, we briefly summarize how we have addressed the reviewers’ concerns and questions.

**Resource Overhead (Reviewers HFN8, G2mv, AmKz, x1y8).** Reviewers raised concerns about the additional costs of deploying Fed-PLoRA, particularly in communication, computation, and memory. We have expanded our discussions of resource overhead in Section 3.5 and provide detailed numerical results in Appendix F.2.

- Communication Overhead (Reviewers HFN8, G2mv, AmKz, x1y8). Fed-PLoRA incurs only limited communication overhead: 0.50 MB per round per client on BERT-base. When the model scales to Llama-3.1-8B, the overhead remains small at 7.09 MB, and even for Qwen3-235B-A22B-Instruct-2507, it reaches only 21.97 MB. These small communication requirements make Fed-PLoRA deployable in low-bandwidth environments.

- Computation Overhead (Reviewer x1y8). The dominant computation cost arises from local training, which remains the same for HETLoRA, FLoRA, FLexLoRA, and Fed-PLoRA, and is on the order of $10^{14}$ FLOPs on BERT-base model. Our method adds only negligible overhead during local initialization (on the order of $10^{7}$ FLOPs), and reduces aggregation cost compared to prior methods, thereby exhibiting strong scalability.

- Memory Overhead (Reviewer HFN8). Fed-PLoRA increases memory usage only through a small temporary buffer that stores additional $R-r_i$ one-rank PLoRA modules during local initialization. For BERT-base model, our method requires only 0.50 MB of temporary client memory overhead, which is negligible compared to the base model’s memory footprint (220.00 MB) and the memory used during local training (2717.91 MB).

**Evaluation on More Models and Benchmarks (Reviewer HFN8).** Following the reviewer’s recommendation, we additionally evaluate on the MATH benchmark with QWen-3, which tests algebra, geometry, number theory, combinatorics, and multi-step symbolic reasoning. We include the results in Appendix D.4 of the revised manuscript.

**Clarification on Experimental Specifications (Reviewer x1y8).** We address the reviewer's questions regarding settings for data distribution, resource heterogeneity configuration, model setup, and ablation study.

**Others.** We report the performance of the untuned base model on all evaluated tasks in Section 4 and Appendix D. We add pseudocode for Fed-PLoRA in Appendix A and discuss sparse LoRA tuning in Appendix B. We also provide implementation code in the supplementary materials.

Overall, we substantially strengthened the manuscript by adding new overhead analyses, multiple new results, clearer experimental specifications, and other discussions. We believe that all major concerns raised by the reviewers have been fully addressed in the revised version, and that the current manuscript presents a more rigorous, complete, and well-justified heterogeneous federated fine-tuning framework.

We hope this consolidated response helps with the final assessment.

Best Regards,

Authors

---

### Meta-Review · Area_Chair_wWqS · 2026-01-05

**Summary:**

The paper proposes a framework for heterogeneous federated fine-tuning (FFT) of large language models. The key idea is to decompose each LoRA module into multiple parallel one-rank modules (PLoRA) and introduce a Select-N-Fold strategy that folds untrained modules into the frozen backbone to mitigate initialization and aggregation noise. Before the rebuttal, this work has four reviewers with three positive reviewers (6, 6, 6) and a negative reviewer (4). During the rebuttal, a positive review raises the score, and another positive review claimed that the rebuttal has addressed most of the concerns. Hence, I agree with three positive reviewers to accept this work.

**Reviewer Concerns:**

Most of concerns have been addressed by the rebuttal. The authors are suggested to prepare the final version based on the rebuttal.

**Reviewer Scores:**

This work has four reviewers with three positive reviewers (6, 6, 6) and a negative reviewer (4). During the rebuttal, a positive review raises the score, and another positive review claimed that the rebuttal has addressed most of the concerns.

---

### Decision · Program_Chairs · 2026-01-26

Accept (Poster)